# Divide and Translate: Compositional First-Order Logic Translation and Verification for Complex Logical Reasoning

**Hyun Ryu**[1]  **Gyeongman Kim**[1]  **Hyemin S. Lee**[2]  **Eunho Yang**[1]
[1]KAIST  [2]MIT
{ryuhyun1905,gmkim,eunhoy}@kaist.ac.kr, hmstella@mit.edu

## Abstract

Complex logical reasoning tasks require a long sequence of reasoning, which a large language model (LLM) with chain-of-thought prompting still falls short. To alleviate this issue, neurosymbolic approaches incorporate a symbolic solver. Specifically, an LLM only translates a natural language problem into a satisfiability (SAT) problem that consists of first-order logic formulas, and a sound symbolic solver returns a mathematically correct solution. However, we discover that LLMs have difficulties to capture complex logical semantics hidden in the natural language during translation. To resolve this limitation, we propose a *Compositional First-Order Logic Translation*. An LLM first parses a natural language sentence into newly defined logical dependency structures that consist of an atomic subsentence and its dependents, then sequentially translate the parsed subsentences. Since multiple logical dependency structures and sequential translations are possible for a single sentence, we also introduce two *Verification* algorithms to ensure more reliable results. We utilize an SAT solver to rigorously compare semantics of generated first-order logic formulas and select the most probable one. We evaluate the proposed method, dubbed CLOVER, on seven logical reasoning benchmarks and show that it outperforms the previous neurosymbolic approaches and achieves new state-of-the-art results.[1]

## 1 Introduction

Logical reasoning involves reaching conclusions through a structured process. It entails drawing inferences by converting information provided in a set of premises into a final conclusion (Nunes, 2012; Bronkhorst et al., 2020). Logical reasoning ability is one of the most challenging metrics to measure intelligence. As a model size grows exponentially, large language models (LLMs) (Brown et al., 2020; Chen et al., 2021; Thoppilan et al., 2022) unlock the ability of machine to reason.

Chain-of-thought (CoT) prompting (Wei et al., 2022) significantly improve the performance of LLMs on simple logical reasoning tasks that require few forward reasoning steps. However, CoT falls short in complex logical reasoning tasks which need longer sequence of reasoning (Ye et al., 2024; Pan et al., 2023). To resolve this issue, several neurosymbolic approaches (Ye et al., 2024; Pan et al., 2023; Kirtania et al., 2024; Olausson et al., 2023) utilize an LLM with a symbolic solver (e.g., an SAT solver) on these complex logical reasoning tasks by the following two steps: 1) an LLM translates the natural language logical reasoning problem into a set of first-order logic formulas, 2) a symbolic solver automatically plans the reasoning steps and executes those to predict an answer of the logical reasoning problem. These approaches take advantages by considering an LLM only as a semantic parser (i.e., a first-order logic translator), which can avoid planning and execution errors by using a symbolic solver.

However, we have discovered that LLMs still cannot translate sentences that represent complex first-order logic. Our experimental evidence in Fig. 1a presents the drastic performance drop of the previous work (Pan et al., 2023) on complex first-order logic translation.[2] The result indicates that an

---

[1]The source code used in the paper is available at `https://github.com/Hyun-Ryu/clover`.

[2]To evaluate the complexity and performance of each first-order logic formula, we sample the first problem from each set of problems that share the same context in the AR-LSAT test set and manually annotate the

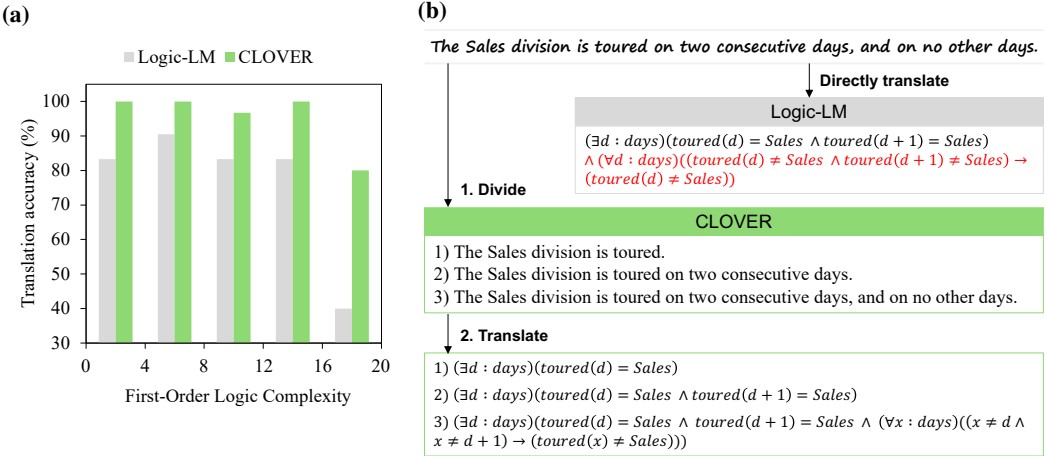

Figure 1: Comparison of a first-order logic translation of the proposed CLOVER and Logic-LM with `gpt-4o` on AR-LSAT. (a) Translation accuracy at different levels of first-order logic complexity. We group the formulas in one of five complexity ranges and report the averaged performance for each range. (b) A representative example. Given declarations of sorts and functions, each method translates the natural language sentence into the corresponding first-order logic formula. We colorize incorrect translation as red for visualization purpose.

LLM performs first-order logic translation faithfully to a certain degree of complexity but falls short beyond that limit. A representative example in Fig. 1b presents an incorrect output of the previous work (Pan et al., 2023) on complex first-order logic translation. The task is to translate a sentence "*The Sales division is toured on two consecutive days, and on no other days.*" into a corresponding first-order logic formula given declarations. An LLM correctly translates a natural language clause "*The Sales division is toured on two consecutive days.*" into a subformula $(\exists d : days) \, (toured(d) = Sales) \land (toured(d+1) = Sales)$ which contains simple logic, but fails to translate "*and on no other days*" which represents more complex logic. Specifically, the incorrectly translated subformula is always true, which has no semantic meaning. After further extensive qualitative error analysis (Appendix K), we conclude that LLMs show promising performance on simple first-order logic translations but does not on complex ones, and the reason is that LLMs have difficulties to discover complex logical structures hidden behind the natural language.

To resolve this limitation, we take a hint from how humans perceive a complex logical sentence to their mind and how they translate it to a first-order logic formula. Since it is hard to immediately comprehend the semantics of a complex logical sentence, humans first understand the semantics of a simpler subsentence and then understand the whole (Montague et al., 1970; Frazier & Fodor, 1978; Sweller, 1988). Inspired by this observation, we use LLM to find the atomic subsentence that does not contain any complex logic and understand other sentence components as dependents of the atomic subsentence. Then, starting with the atomic subsentence, we use LLM to translate subsentences by accumulating sentence components. This could help LLM to preserve first-order logic semantics during translation. To rigorously define the atomic subsentence and sentence components with logical meaning, we introduce a new parsing method for natural language that represents first-order logic, called *logical dependency parsing* (Section 3.1).

Based on logical dependency parsing, we propose a compositional first-order logic translation (Section 3.2) by few-shot learning with an LLM. It consists of the following three steps: logical dependency parsing, component accumulation, and sequential translation. First, a target sentence is parsed into logical dependency structures which consist of components of the sentence and their logical dependencies. Second, components are accumulated while preserving their logical dependencies, where the last accumulated sentence is the target sentence. Finally, each accumulated sentence is sequentially translated into first-order logic formula in the order of accumulations, where the last formula is an estimated formula of the target sentence.

Not only that, since there could be multiple outputs on logical dependency parsing and sequential translation, we introduce verification algorithms to ensure more reliable first-order logic transla-

---

ground truth formulas. Detailed information of measuring complexity is in Appendix B and the process of the annotated subset construction is described in Appendix F.

tion (Section 3.3). We propose two verification algorithms: logical consistency and disproving by counter-interpretation. To fully leverage the deterministic nature of first-order logic, we use an SAT solver to compare any two formulas. Logical consistency selects the most frequent logically equivalent formulas. However, we observe that an LLM sometimes make logically consistent translation errors. To overcome such a limitation, we devise disproving by counter-interpretation. It sequentially compares two formulas and disprove one of them by determining if a counter-interpretation to equivalence of two formulas satisfies the target sentence. The last formula remained is then selected. To save computational cost, we compare each one of logically equivalent formulas.

We evaluate the proposed CLOVER, a **C**ompositional First-Order **Lo**gic Translation and **Ver**ification, on seven logical reasoning benchmarks (Section 4). CLOVER outperforms the previous neurosymbolic approaches and achieves the new state-of-the-art performance. It also significantly enhances the first-order logic translation accuracy across all levels of complexity and the largest performance gain occurs at the highest first-order logic complexity (Fig. 1a).

To summarize our contributions,

1. We introduce CLOVER, a novel neurosymbolic approach that enhances complex logical reasoning in LLMs by compositional translation of natural language into first-order logic and verification of logical semantics.

2. We newly define a *logical dependency structure* to decompose logical sentences while preserving an underlying first-order logic semantics.

3. We also propose two SAT-based first-order logic verification algorithms that can faithfully select a correctly translated formula.

4. We evaluate CLOVER on seven logical reasoning benchmarks and show that CLOVER outperforms the previous neurosymbolic apporoaches and achieves the new state-of-the-art performance.

## 2 PROBLEM FORMULATION

Through the lens of (many-sorted) first-order logic[3], a logical reasoning problem $x$ is a natural language description of a $\Sigma$-*theory* $\mathcal{T}$[4], constraints $\Phi$, and a query $q$, denoted as $x = NL(\mathcal{T}, \Phi, q)$. A $\Sigma$-*theory* $\mathcal{T}$ is a non-empty set of any $\Sigma$-*structure* where a *signature* $\Sigma = (S, F, P)$ consists of *sorts* $S$, *function symbols* $F$, and *predicate symbols* $P$. Hereinafter, we denote the vocabulary of first-order logic as *italic* for clarity, and omit the prefix "$\Sigma$-" for simplicity. A *structure* of a *theory* indicates the semantics of *formulas*. Constraints $\Phi$ are a set of *formulas* that are true, denoted as $\Phi = \{\phi_1, \phi_2, \cdots, \phi_K\}$. A query $q$ is also a *formula* which is yet determined as true, false, or unknown given the constraints $\Phi$.

**Prior works.** Prior neurosymbolic approaches (Ye et al., 2024; Pan et al., 2023; Kirtania et al., 2024; Olausson et al., 2023) directly translate the logical reasoning problem $x$ into a set of first-order logic using an LLM and then employ a symbolic solver (e.g., an SAT solver) to solve an SAT problem. In these methods, an LLM performs a single inference for the first-order logic translation as follows:

$$\hat{\mathcal{T}}, \{\hat{\varphi}_k, \hat{NL}(\varphi_k)\}_{k=1}^{K+1} \sim P_{\text{LLM}}(\mathcal{T}, \{\varphi_k, NL(\varphi_k)\}_{k=1}^{K+1} \mid x, \mathbf{x}_{\text{fs}}) \tag{1}$$

where $\varphi_k = \phi_k$ for $1 \leq k \leq K$ and $\varphi_{K+1} = q$, and a few-shot exemplar set $\mathbf{x}_{\text{fs}} = \{x^{(i)}, \mathcal{T}^{(i)}, \{\varphi_k^{(i)}, NL(\varphi_k^{(i)})\}_{k=1}^{K^{(i)}+1}\}_{i=1}^N$ with the size of the set $N$. However, it often generates more than one *formulas* for a single target sentence or generates a *formula* which is a translation of combination of a target sentence and part of other sentences. Though it might be logically correct as a whole, we cannot further analyze and verify the translation at a sentence-level.

**First-order logic translation.** To resolve this drawback, we perform first-order logic translation for each sentence. Since sentence-level translations require a pre-defined *theory* and target sentences, we first generate a theory $\hat{\mathcal{T}}$ and a set of natural language sentences $\{\hat{NL}(\varphi_k)\}_{k=1}^{K+1}$ from

---

[3]Many-sorted first-order logic is one of the variants of the standard first-order logic that allows variables to have different domains, which is called sorts $S$. We provide related preliminaries in Appendix A.

[4]A *theory* assigns specific meanings to symbols of *formulas*. For simplicity, we presume that a *theory* $\mathcal{T}$ incorporates the most commonly applied theories (e.g., theory of equality, arithmetic, etc.).

$x$, and then generate $\hat{\mathcal{T}}$-*satisfiable formula* $\hat{\varphi}_k$ for each sentence $\hat{NL}(\varphi_k)$. To be specific, the single inference by the LLM in Eq. 1 is separated into the following two steps: 1) given a logical reasoning problem $x$ and a few-shot exemplar set $\mathbf{x}_{\text{fs}}^{\text{prep}} = \{x^{(i)}, \mathcal{T}^{(i)}, \{NL(\varphi_k^{(i)})\}_{k=1}^{K^{(i)}+1}\}_{i=1}^N$, the LLM generates a tuple of an estimated theory and a set of natural language sentences, denoted $x^{\text{prep}} = (\hat{\mathcal{T}}, \{\hat{NL}(\varphi_k)\}_{k=1}^{K+1})$, 2) given the theory $\hat{\mathcal{T}}$ and a set of few-shot exemplar sets $\mathbf{X}_{\text{fs}}$, the proposed CLOVER translates each natural language sentence $\hat{NL}(\varphi_k)$ into the estimated *formula* $\hat{\varphi}_k$ that is $\hat{\mathcal{T}}$-*satisfiable* as follows:

$$\hat{\mathcal{T}}, \{\hat{NL}(\varphi_k)\}_{k=1}^{K+1} \sim P_{\text{LLM}}(\mathcal{T}, \{NL(\varphi_k)\}_{k=1}^{K+1} \mid x, \mathbf{x}_{\text{fs}}^{\text{prep}})$$
$$\hat{\varphi}_k = \text{CLOVER}(\hat{\mathcal{T}}, \hat{NL}(\varphi_k), \mathbf{X}_{\text{fs}}), \forall k \in \{1, 2, \cdots, K+1\}. \tag{2}$$

A detailed description of the set of few-shot exemplar sets $\mathbf{X}_{\text{fs}} = \{\mathbf{x}_{\text{fs}}^{\text{parse}}, \mathbf{x}_{\text{fs}}^{\text{accum}}, \mathbf{x}_{\text{fs}}^{\text{trans}}, \mathbf{x}_{\text{fs}}^{\text{disprv}}\}$ and the proposed CLOVER for $x^{\text{prep}} = (\hat{\mathcal{T}}, \{\hat{NL}(\varphi_k)\}_{k=1}^{K+1})$ will be discussed in the following section.

**SAT problem solving.** Once estimations of the theory $\hat{\mathcal{T}}$, constraints $\hat{\Phi} = \{\hat{\varphi}_1, \hat{\varphi}_2, \cdots, \hat{\varphi}_K\}$, and a query $\hat{q} = \hat{\varphi}_{K+1}$ are completed for the logical reasoning problem $x$, these form an SAT problem $\mathcal{P} = (\hat{\mathcal{T}}, \hat{\Phi}, \hat{q})$. An automated SAT solver then determines the $\hat{\mathcal{T}}$-*satisfiability*[5] of the query $\hat{q}$ under the constraints $\hat{\Phi}$, which is a final prediction of an answer of the logical reasoning problem $x$. We use a Z3 theorem prover (De Moura & Bjørner, 2008) as an SAT solver in the implementation.

## 3 CLOVER

In this section, we propose CLOVER, a **C**ompositional First-Order **Lo**gic Translation and **Ver**ification for complex logical reasoning. To fully capture first-order logic semantics in natural language, it first parses a single natural language sentence into logical dependency structures. Then, it sequentially translates parsed subsentences with an LLM. Since there are multiple ways to parse and translate the sentences, we also introduce two SAT-based verification algorithms to thoroughly compare semantics of translated first-order logic formulas.

### 3.1 LOGICAL DEPENDENCY STRUCTURES

Logical dependency structure $\mathcal{A}$ of a sentence $NL(\varphi)$ under the theory $\mathcal{T}$ where $\varphi$ is $\mathcal{T}$-*satisfiable* is defined by components and their logical dependencies. First, components are natural language building blocks of logical dependency structures of a sentence, which consist of logic units $U$, logic couplers $C$, and logic dependents $D$. The following definitions formally describe each of them.

**Definition 1** (Logic units). Given a sentence $NL(\varphi)$ and a theory $\mathcal{T}$ where $\varphi$ is $\mathcal{T}$-*satisfiable*, logic units $U$ are the natural language descriptions of an *atom* of $\varphi$.

**Definition 2** (Logic couplers). Logic couplers $C$ are either conjunctions or an operator named merge. Merge combines two logic units which contain the natural language describing the same *term* without adding any conjunction.

**Definition 3** (Logic dependents). Logic dependents $D$ are components neither logic units nor logic couplers which logically depend on another component.

Second, we define logical dependency between two components, and the following definition formally describes it.

**Definition 4** (Logical dependency). The component $X$ is said to logically depend on the component $Y$ in the given sentence if and only if the meaning of $Y$ is (or includes) a predicate and the meaning of $X$ is an argument of this predicate in the sentence.

We also introduce properties of logical dependency structure stemmed from its definition.

*Remark* 1. A given sentence and theory can have multiple logical dependency structures.

*Remark* 2. All components except for one should logically depend on another component.

*Remark* 3. No logic dependent logically depends on a logic coupler.

We present examples of logical dependency structures in Fig. 2 and in Appendix D.

---

[5]For AR-LSAT, we need to check the $\hat{\mathcal{T}}$-*validity* depending on the problem. More details in Appendix E.

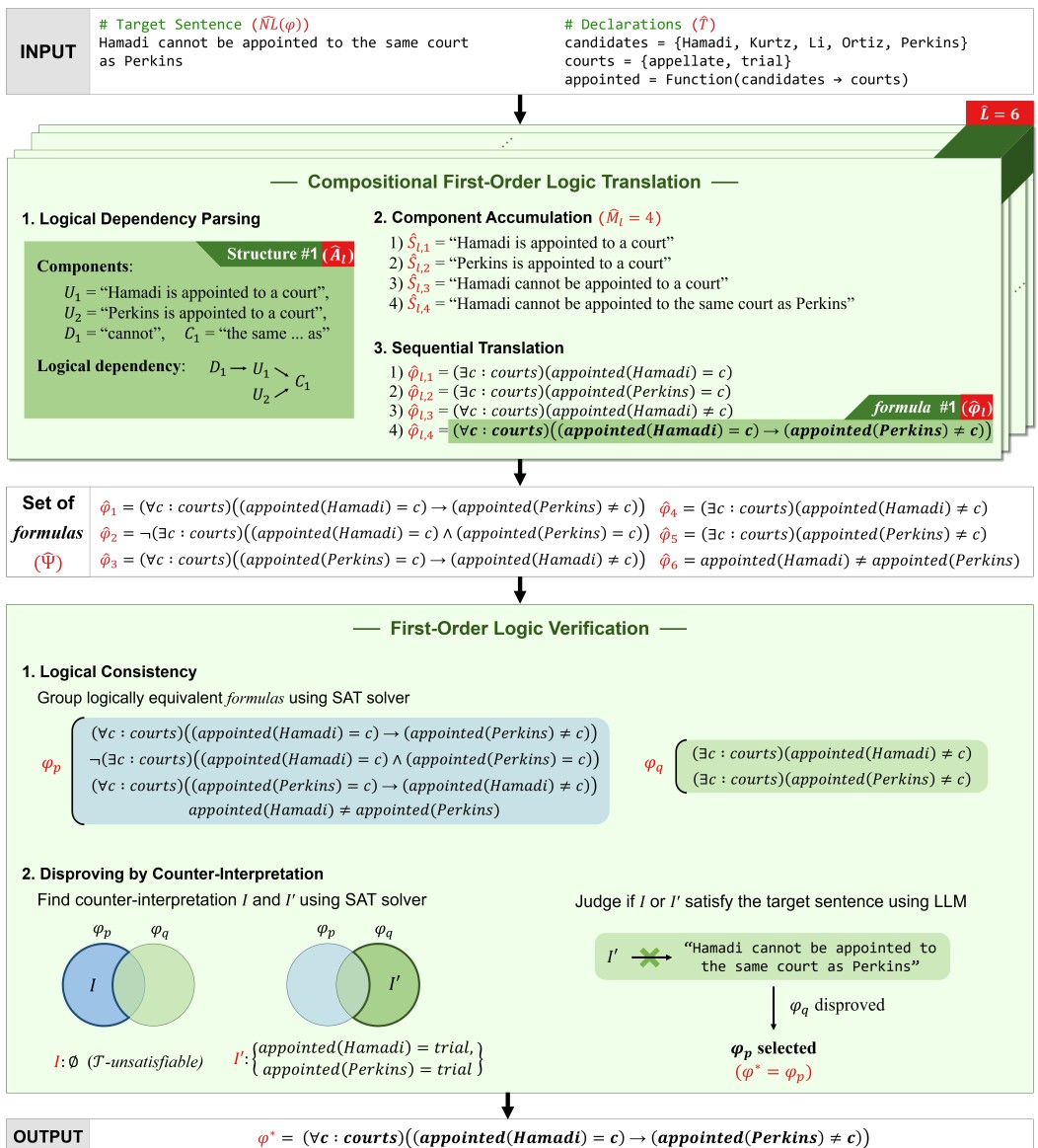

Figure 2: Overview of CLOVER. Given declarations of a theory, CLOVER parses a target sentence to several possible logical dependency structures, accumulates components according to logical dependencies, and sequentially translates subsentences to first-order logic *formulas*. Then, CLOVER verifies a set of estimated formulas. Logical consistency selects the most frequent logically equivalent formulas. Disproving by counter-interpretation sequentially compares two formulas and disprove one by determining if a counter-interpretation satisfies the target sentence.

## 3.2 COMPOSITIONAL FIRST-ORDER LOGIC TRANSLATION

To compositionally translate natural language sentences to first-order logic *formulas* under given theory for $x^{\text{prep}} = (\hat{\mathcal{T}}, \{\hat{NL}(\varphi_k)\}_{k=1}^{K+1})$, we adhere to the following three steps by few-shot learning with an LLM. We describe the following steps for a single target sentence $\hat{NL}(\varphi)$ of a *formula* $\varphi \in \{\varphi_k\}_{k=1}^{K+1}$.

**Logical Dependency Parsing.** In the first step, a target sentence is parsed into different possible logical dependency structures. An LLM is given a definition of logical dependency structures (Section 3.1), a target sentence $\hat{NL}(\varphi)$ and its theory $\hat{\mathcal{T}}$, and a few-shot exemplar set $\mathbf{x}_{\text{fs}}^{\text{parse}} = \{\mathcal{T}^{(i)}, NL(\varphi^{(i)}), \{\mathcal{A}_l^{(i)}\}_{l=1}^{L^{(i)}}\}_{i=1}^{N}$ for logical dependency parsing. $L^{(i)}$ is a size of a set

of different possible logical dependency structures of a sentence $NL(\varphi^{(i)})$ under the theory $\mathcal{T}^{(i)}$, and $N$ is a size of the few-shot exemplar set. Then, LLM generates a set of different possible logical dependency structures $\{\hat{\mathcal{A}}_l\}_{l=1}^{\hat{L}}$ of the target sentence with the size of the set $\hat{L}$ as follows:

$$\hat{L}, \{\hat{\mathcal{A}}_l\}_{l=1}^{\hat{L}} \sim P_{\text{LLM}}(L, \{\mathcal{A}_l\}_{l=1}^{L} \mid \hat{\mathcal{T}}, \hat{NL}(\varphi), \mathbf{x}_{\text{fs}}^{\text{parse}}). \tag{3}$$

**Component Accumulation.** In the second step, components of a logical dependency structure are accumulated to gradually compose new sentences until those reach the target sentence. We present the rules for component accumulation in Appendix C. An LLM is given a definition of logical dependency structures (Section 3.1), rules for component accumulation (Appendix C), a target sentence $\hat{NL}(\varphi)$ and one of its logical dependency structures $\hat{\mathcal{A}}_l$ where $l \in \{1, 2, \cdots, \hat{L}\}$, and a few-shot exemplar set $\mathbf{x}_{\text{fs}}^{\text{accum}} = \{NL(\varphi^{(i)}), \mathcal{A}^{(i)}, (S_m^{(i)})_{m=1}^{M^{(i)}}\}_{i=1}^{N}$ for component accumulation. $(S_m^{(i)})_{m=1}^{M^{(i)}}$ is a sequence of accumulated sentences where $M^{(i)}$ is the length of the sequence. Then, LLM generates a sequence of sentences $(\hat{S}_{l,m})_{m=1}^{\hat{M}_l}$ where $\hat{M}_l$ is the length of the estimated sequence as follows:

$$\hat{M}_l, (\hat{S}_{l,m})_{m=1}^{\hat{M}_l} \sim P_{\text{LLM}}(M_l, (S_{l,m})_{m=1}^{M_l} \mid \hat{NL}(\varphi), \hat{\mathcal{A}}_l, \mathbf{x}_{\text{fs}}^{\text{accum}}), \forall l \in \{1, 2, \cdots, \hat{L}\}. \tag{4}$$

The last sentence of accumulation $\hat{S}_{l,\hat{M}_l}$ is the target sentence $\hat{NL}(\varphi)$. We present examples of component accumulation in Appendix D.

**Sequential Translation.** In the last step, accumulated natural language sentences are sequentially translated into first-order logic *formulas*, which the target sentence is finally translated. An LLM is given a sequence of accumulated sentences $(\hat{S}_{l,m})_{m=1}^{\hat{M}_l}$ of a target sentence where $l \in \{1, 2, \cdots, \hat{L}\}$, a theory $\hat{\mathcal{T}}$, and a few-shot exemplar set $\mathbf{x}_{\text{fs}}^{\text{trans}} = \{\mathcal{T}^{(i)}, (S_m^{(i)})_{m=1}^{M^{(i)}}, (\varphi_m^{(i)})_{m=1}^{M^{(i)}}\}_{i=1}^{N}$ for first-order logic translation. Then, LLM generates a sequence of *formulas* $(\hat{\varphi}_{l,m})_{m=1}^{\hat{M}_l}$ as follows:

$$(\hat{\varphi}_{l,m})_{m=1}^{\hat{M}_l} \sim P_{\text{LLM}}((\varphi_{l,m})_{m=1}^{\hat{M}_l} \mid \hat{\mathcal{T}}, (\hat{S}_{l,m})_{m=1}^{\hat{M}_l}, \mathbf{x}_{\text{fs}}^{\text{trans}}), \forall l \in \{1, 2, \cdots, \hat{L}\}. \tag{5}$$

The last *formula* of the sequence is the first-order logic translation of the target sentence (i.e., $\hat{\varphi}_l = \hat{\varphi}_{l,\hat{M}_l}$). For $\forall l \in \{1, 2, \cdots, \hat{L}\}$, we could generate a set of estimated *formulas* $\hat{\Psi} = \{\hat{\varphi}_l\}_{l=1}^{\hat{L}}$ for a target sentence $\hat{NL}(\varphi)$. In practice, we randomly sample multiple times to enrich the pool of estimated *formulas* that benefits the second stage of CLOVER, first-order logic verification.

## 3.3 FIRST-ORDER LOGIC VERIFICATION

To select the most probable *formula* in a set of compositionally translated first-order logic *formulas* $\hat{\Psi}$, we introduce the following two algorithms using an SAT solver (and few-shot learning with an LLM). As in Section 3.2, we describe the following algorithms for a single target sentence $\hat{NL}(\varphi)$ under the theory $\hat{\mathcal{T}}$ (i.e., The algorithms select a verified *formula* $\varphi^*$ in a set of estimated *formulas* $\hat{\Psi}$). Prior to describing the detailed algorithms, we filter out the *formulas* that are syntactically incorrect or $\hat{\mathcal{T}}$-*unsatisfiable* in $\hat{\Psi}$ using an SAT solver and call the processed set $\hat{\Psi}_{\text{sat}}$.

**Logical Consistency.** We select the most frequent logically equivalent *formulas*, which we call this algorithm logical consistency. It presumes an LLM utilizes different logical dependency structures to generate several *formulas* that are logically equivalent. An LLM might also make mistake in intermediate steps of compositional first-order logic translation and generate incorrect *formulas*, but these are less likely to be logically equivalent. For each pair of *formulas* $(\varphi_p, \varphi_q)$ such that $\varphi_p \in \hat{\Psi}_{\text{sat}}, \varphi_q \in \hat{\Psi}_{\text{sat}}$, and $p \neq q$, an SAT solver determines their $\hat{\mathcal{T}}$-*equivalence*. Then, we group $\hat{\mathcal{T}}$-*equivalent formulas* and select any *formula* in the group that has the largest number of elements. However, we observe that an LLM sometimes makes consistent mistakes in the last step of compositional first-order logic translation, which leads to logically equivalent incorrect *formulas*.

**Disproving by Counter-Interpretation.** To resolve this issue, we introduce an advanced algorithm that sequentially disproves incorrect *formulas* by *counter-interpretation*. Following this al-

---

**Algorithm 1** First-Order Logic Verification (Disproving by Counter-Interpretation)

---

**Input**: Theory $\mathcal{T}$, a natural language sentence $NL(\varphi)$ of a first-order logic *formula* $\varphi$, and a set of estimated $\mathcal{T}$-*satisfiable formulas* $\hat{\Psi}_{\text{sat}}$
**Output**: Verified *formula* $\varphi^*$

$\hat{\varphi}_0 \sim \hat{\Psi}_{sat}$         $\triangleright$ *Select an element $\hat{\varphi}_0$ in $\hat{\Psi}_{sat}$ randomly*
$\varphi^* \leftarrow \hat{\varphi}_0$         $\triangleright$ *Initialize $\varphi^*$ to a random element $\hat{\varphi}_0$*
**for** each $\hat{\varphi} \in \hat{\Psi}_{sat} \setminus \{\hat{\varphi}_0\}$ **do**
   $\varphi_{\text{temp}} \leftarrow \varphi^*$         $\triangleright$ *Use a temporary variable $\varphi_{temp}$ for the update*
   **for** each $(\varphi_p, \varphi_q) \in \{(\varphi^*, \hat{\varphi}), (\hat{\varphi}, \varphi^*)\}$ **do**
      **if** $(\varphi_p \wedge \neg\varphi_q)$ is $\mathcal{T}$-*satisfiable* **then**
         find $\mathcal{T}$-*interpretation* $I$ such that $I \vDash (\varphi_p \wedge \neg\varphi_q)$    $\triangleright$ *SAT solver finds $I$ if it exists*
         $\hat{e} \sim P_{LLM}(e \mid NL(\varphi), I, \mathbf{x}_{fs}^{disprv}), e \in \{\top, \bot\}$    $\triangleright$ *LLM determines if $I \vDash \varphi$*
         **if** $(\varphi_p = \varphi^* \wedge \neg\hat{e})$ **or** $(\varphi_p = \hat{\varphi} \wedge \hat{e})$ **then**
            $\varphi_{\text{temp}} \leftarrow \hat{\varphi}$
         **end if**
      **end if**
   **end for**
   $\varphi^* \leftarrow \varphi_{\text{temp}}$         $\triangleright$ *Update $\varphi^*$ after checking $\mathcal{T}$-interpretations from both side*
**end for**
**return** $\varphi^*$

---

gorithm, an accurate *formula* remains the last if it exists in $\hat{\Psi}_{\text{sat}}$. Specifically, we select a random element $\hat{\varphi}_0$ in $\hat{\Psi}_{\text{sat}}$ and initialize the verified *formula* $\varphi^*$ to $\hat{\varphi}_0$. For each estimated *formula* $\hat{\varphi}$ in $\hat{\Psi}_{\text{sat}} \setminus \{\hat{\varphi}_0\}$, an SAT solver determines if $(\varphi^* \wedge \neg\hat{\varphi})$ is $\hat{\mathcal{T}}$-*satisfiable*. First, if it is $\hat{\mathcal{T}}$-*satisfiable*, an SAT solver finds a *counter-interpretation* $I$ to a $\hat{\mathcal{T}}$-*equivalence* of $\varphi^*$ and $\hat{\varphi}$ that satisfies $(\varphi^* \wedge \neg\hat{\varphi})$. Given a target sentence $\hat{NL}(\varphi)$, a *counter-interpretation* $I$, and a few-shot exemplar set $\mathbf{x}_{\text{fs}}^{\text{disprv}} = \{NL(\varphi)^{(i)}, I^{(i)}, e^{(i)}\}_{i=1}^{N}$ for disproving, an LLM decides if $I$ *satisfies* $\varphi$, which returns a boolean value $\hat{e}$. If $\hat{e}$ is True, then $\hat{\varphi}$ is disproved since $I$ does not *satisfy* $\hat{\varphi}$ but *satisfies* $\varphi$. If $\hat{e}$ is False, then $\varphi^*$ is disproved since $I$ *satisfies* $\varphi^*$ but does not *satisfy* $\varphi$. Second, if $(\varphi^* \wedge \neg\hat{\varphi})$ is $\hat{\mathcal{T}}$-*unsatisfiable*, it is equivalent to $(\varphi^* \rightarrow \hat{\varphi})$ is $\hat{\mathcal{T}}$-*satisfiable*, and no $I$ exists. After repeating this decision process for $(\hat{\varphi} \wedge \neg\varphi^*)$, we can consider a *counter-interpretation* $I$ that satisfies $(\hat{\varphi} \wedge \neg\varphi^*)$ and disproves accordingly. We select the verified *formula* $\varphi^*$ that remains the last. Algorithm 1 summarizes the whole process.

## 4 EXPERIMENTS

### 4.1 SETUP

**Tasks.** We evaluate CLOVER on seven logical reasoning tasks: AR-LSAT (Zhong et al., 2022), ZebraLogic (Lin et al., 2025), Logic grid puzzle (Puzzle), Symbol interpretation (Symbol), and Logical deduction (Deduction) from the BigBench collaborative benchmark (Srivastava et al., 2022), FOLIO (Han et al., 2022), and ProofWriter (Tafjord et al., 2021). AR-LSAT consists of analytical reasoning problems of the law school admission test, and ZebraLogic is a benchmark for zebra puzzles. Puzzle, Symbol, and Deduction are tasks from logical reasoning category in the BigBench. FOLIO[6] is an expert-written first-order logic reasoning task, and ProofWriter is a deductive reasoning benchmark. Note that all tasks except ZebraLogic are multiple choice problems, and Appendix F describes details of each task.

**Language Models.** We perform our experiments mainly on `gpt-4o` (Achiam et al., 2023), a current state-of-the-art LLM for complex, multi-step tasks, unless stated. We also evaluate CLOVER

---

[6]We use a revised version of FOLIO that improves sample quality and fixes errors, which is released on: `https://huggingface.co/datasets/yale-nlp/FOLIO`.

Table 1: Performance on logical reasoning tasks using CLOVER and the baseline methods.

|  | AR-LSAT | ZebraLogic | Puzzle | Symbol | Deduction | FOLIO | ProofWriter |
|---|---|---|---|---|---|---|---|
| Standard | 30.3 | 0.4 | 63.0 | 74.7 | 84.7 | 70.9 | 53.7 |
| CoT | 36.8 | 0.4 | 51.0 | 80.8 | 94.0 | 73.9 | 78.0 |
| SymbCoT | 34.2 | 0.8 | 66.5 | 55.6 | 90.7 | 76.9 | 80.2 |
| Logic-LM | 42.4 | 45.4 | 64.0 | 81.8 | 95.3 | 75.4 | 95.3 |
| CLOVER | **62.8** | **75.4** | **83.5** | **89.9** | **99.3** | **78.8** | **96.7** |

Table 2: Comparison of program accuracy, execution rate, and execution accuracy of CLOVER and Logic-LM.

|  | Program Acc | | Execution Rate | | Execution Acc | |
|---|---|---|---|---|---|---|
|  | Logic-LM | CLOVER | Logic-LM | CLOVER | Logic-LM | CLOVER |
| AR-LSAT | 17.3 | **46.8** | 33.8 | **59.7** | 51.3 | **78.3** |
| Puzzle | 60.0 | **79.0** | 79.5 | **80.0** | 75.5 | **98.8** |
| Symbol | 49.5 | **76.8** | 52.5 | **82.8** | **94.2** | 92.7 |
| Deduction | 92.7 | **99.0** | 97.3 | **99.7** | 95.2 | **99.3** |
| FOLIO | 51.2 | **62.6** | 65.5 | **74.9** | 78.2 | **83.6** |
| ProofWrtier | 94.2 | **96.5** | 96.8 | **99.2** | 97.2 | **97.3** |

and the baselines using a smaller model, `gpt-4o-mini` (Achiam et al., 2023).[7] To reproduce our experiments, we set the temperature to 0 and select the highest probability response from the model.

**Baselines.** We compare CLOVER primarily to Logic-LM (Pan et al., 2023), a state-of-the-art neurosymbolic approach for logical reasoning. There are few more works (Ye et al., 2024; Olausson et al., 2023) nearly the same to Logic-LM, but we focus on Logic-LM since their difference is marginal. We also compare CLOVER to another neurosymbolic approach (Xu et al., 2024) which uses an LLM to solve SAT problems instead of using a symbolic solver. In addition, we compare to the standard prompting and CoT prompting that leverages in-context learning capability of the base LLMs. For fair comparison, we manually sample or derive our few-shot exemplar sets from those in the previous works (Pan et al., 2023; Xu et al., 2024) if it is possible. Since the previous works do not evaluate their models on ZebraLogic, Puzzle, and Symbol, we randomly select a single exemplar problem outside the test set. We demonstrate exemplar few-shot prompts in Appendix J.

**Evaluation metrics.** We measure the performance of CLOVER and the baselines primarily by the correctness of logical reasoning problems. For neurosymbolic approaches with a symbolic solver, if the solver cannot execute the translated SAT problem, we fall back to CoT predictions. From this unique property, following Pan et al. (2023), we use three additional evaluation metrics: program accuracy, execution rate, and execution accuracy, for multiple choice problems. Program accuracy does not include the CoT predictions for unexecutable problems. Execution rate measures the portion of executable problems, and execution accuracy indicates the accuracy for executable problems.

## 4.2 RESULTS

We present the performance of CLOVER and the baselines on different tasks, different evaluation metrics, and different language model scales. First, Table 1 compares the performance of CLOVER and the baselines on seven logical reasoning tasks. CLOVER outperforms Logic-LM and other baselines by a significant margin across different logical reasoning tasks. CLOVER shows marked improvement on hard logical reasoning tasks. Specifically, it enhances the performance of Logic-LM on AR-LSAT by 20.4% and ZebraLogic by 30.0%. Overall, neurosymbolic approaches with a symbolic solver (CLOVER and Logic-LM) show remarkable improvement on these hard reasoning tasks. The inference time costs of CLOVER and the baselines are reported in Appendix H.

---

[7]To specify language model versions provided by OpenAI, we use `gpt-4o-2024-05-13` and `gpt-4o-mini-2024-07-18` on our experiments.

Table 3: Ablation of CLOVER on AR-LSAT and ZebraLogic. The first three rows are ablations of CLOVER, and the last two rows correspond to CLOVER.

| Is CLOVER? | Translation | Verification | AR-LSAT | ZebraLogic |
|:---:|:---:|:---:|:---:|:---:|
| ✗ | direct | ✗ | 53.3 | 45.4 |
| ✗ | direct (5×) | logical consistency | 54.6 | 59.6 |
| ✗ | compositional | ✗ | 55.0 | 70.0 |
| ✓ | compositional | logical consistency | 61.9 | 74.2 |
| ✓ | compositional | disproving | **62.8** | **75.4** |

Second, Table 2 presents three additional evaluations for the neurosymbolic approaches with a symbolic solver. CLOVER shows higher execution rate on every task, which indicates that CLOVER has better capability to generate syntactically correct first-order logic formulas than Logic-LM. CLOVER also shows higher execution accuracy on most tasks, which indicates that CLOVER has better capability to generate logically (or semantically) correct formulas than Logic-LM. These two observations lead to an outperforming program accuracy of CLOVER across different logical reasoning tasks. Specifically, CLOVER increases the execution rate of Logic-LM on AR-LSAT by 25.9% and the execution accuracy by 27.0%, which finally leads to more than doubled program accuracy of Logic-LM. Lastly, we compare the performance of CLOVER and the baselines on different languange models in Appendix G.

## 4.3 ABLATIONS

We conduct ablation studies of CLOVER on two perspectives: compositional translation and verification, in Table 3. Ablating verification from CLOVER (i.e., random selection) shows 6.9% and 4.2% performance degradation on AR-LSAT and ZebraLogic, respectively. It clearly supports the effectiveness of the verification. Ablating compositional translation from CLOVER (i.e., direct translation) shows 7.3% and 14.6% performance degradation on AR-LSAT and ZebraLogic, respectively. It also clearly supports the effectiveness of the compositional translation. To maintain the verification stage as is, we repeat the sampling of direct translation five times, which is slightly larger than the average number of estimated formulas of CLOVER. Ablating both compositional translation and verification from CLOVER shows further performance loss. Additionally, disproving by counter-interpretation yields better performance than logical consistency.

## 4.4 ANALYSIS

**Types of Errors.** We analyze error types of CLOVER on AR-LSAT and compare those to Logic-LM's in Figure 3. Since an SAT solver is *sound* and does not cause any error, our error analysis focuses on the first-order logic translation. Logic-LM's errors are mainly caused by incorrect logic (or semantic) and incorrect syntax, which take 53.7% of the total errors. There are preprocessing errors and other errors caused by an incorrect selection of a satisfiability function and limited expressiveness of a Z3 theorem prover. In contrast, *we highlight that CLOVER has nearly no logic or syntax error*. CLOVER's errors are primarily caused by preprocessing and other errors, which takes 78.6% of the total errors. This analysis indicates that CLOVER significantly enhances the ability of a language model to generate both syntactically and semantically precise first-order logic formulas.

**Robustness on Reasoning Length.** We present robustness of CLOVER on long sequence of reasoning and compare the results with CoT-based reasoning LLMs (Jaech et al., 2024)[8] in Figure 4. We also add the results of Logic-LM to measure the effect of neurosymbolic approach on reasoning length. We observe a noticeable performance drop of CoT-based reasoning LLMs on long sequence of reasoning, which is a frequently pointed-out drawback of the CoT-based approaches. However, neurosymbolic approaches show robustness to the reasoning length. Specifically, CLOVER shows only 12.5% performance drop between the tasks of the shortest and longest sequence of reasoning.

---

[8]We use CoT-based reasoning LLMs that were recently released from OpenAI, specifically `o1-preview-2024-09-12` and `o1-mini-2024-09-12`, for our analysis.

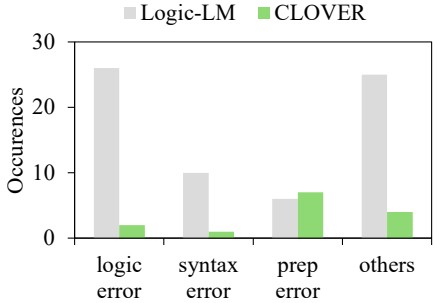

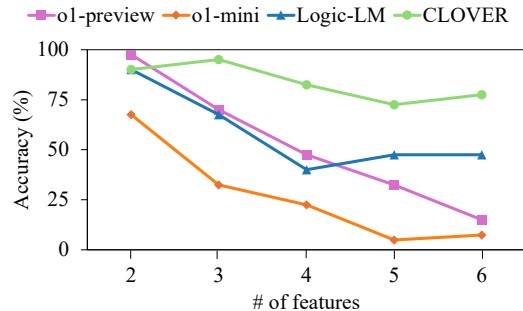

Figure 3: Occurences of different error types of CLOVER and Logic-LM on AR-LSAT annotated subset.

Figure 4: Performance on different reasoning lengths using CoT-based reasoning LLMs and neurosymbolic approaches on ZebraLogic. Each example is a puzzle of five houses with varying # of features.

## 5 RELATED WORKS

**LLM-based neurosymbolic approach for reasoning.** Previous works (Ye et al., 2024; Pan et al., 2023; Olausson et al., 2023; Kirtania et al., 2024) utilize an LLM as a semantic parser which translates the natural language logical reasoning problems into first-order logic formulas, and then use a symbolic solver to automatically solve an SAT problem. There is another work (Xu et al., 2024) that utilizes an LLM as not only a semantic parser but also a symbolic solver and a verifier for semantic parsing and symbolic solving. However, these previous works share a common drawback that an LLM cannot faithfully perform complex first-order logic translation, which fundamentally limits the performance of neurosymbolic approaches on complex logical reasoning tasks.

**LLM-based problem decomposition.** To solve natural language tasks, previous works explore decomposing complex problems into several simpler ones using LLMs. Drozdov et al. (2022) use an LLM to syntactically parse the natural language sentence into several subsentences and performs compositional semantic parsing for simple tasks such as text-to-SQL. However, since a syntactic parsing cannot preserve the semantic of logic, Drozdov et al. (2022) is not applicable to complex logical reasoning tasks. Other works (Zhou et al., 2023; Khot et al., 2023; Press et al., 2023; Dua et al., 2022; Ye et al., 2023) focus on decomposing simple question-answering problems by prompting LLMs with few-shot examples. However, if LLMs simply rely on few-shot examples for decomposing complex logical reasoning problems, then the problems might be incorrectly decomposed, which leads to an unexpected performance loss.

**LLM-generated formal language verification.** There are lines of works to verify formal language generated by LLMs. Chen et al. (2024) and Madaan et al. (2024) first generate a code from natural language, get feedback from an LLM, and refine the code based on the feedback. Chen et al. (2024) additionally utilizes an external feedback signal from an executor. Ni et al. (2023) first generates candidate codes from natural language and then verify by predicting their correctness using a trained neural network. However, these model-based verifications show limited performance on complex logical reasoning tasks (Appendix I).

## 6 CONCLUSION

We propose CLOVER, a compositional first-order logic translation and verification for complex logical reasoning. CLOVER first parses the natural language sentence into newly defined logical dependency structures, which reflect first-order logic semantics hidden in the natural language, and then compositionally translates the sentence. We also introduce two verification algorithms using satisfiability to fully cover first-order logic semantics. Empirical results show that CLOVER achieves state-of-the-art performance on seven logical reasoning benchmarks.

# 7 ACKNOWLEDGMENTS

This work was partly supported by Institute for Information & communications Technology Promotion(IITP) (No.RS-2019-II190075, Artificial Intelligence Graduate School Program(KAIST), No.2022-0-00713, Meta-learning applicable to real-world problems, No.RS-2024-00457882, AI Research Hub Project) and National Research Foundation of Korea (NRF) (No.RS-2023- 00209060, A Study on Optimization and Network Interpretation Method for Large-Scale Machine Learning) grant funded by the Korea government (MSIT).

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

## A  FIRST-ORDER LOGIC PRELIMINARIES

To clarify the first-order logic vocabularies used in the paper, we give basic definitions as preliminaries based on Ranise et al. (2005) and Jovanović & Barrett (2011). To help understanding, we also provide an example of the definitions, where the example is sampled from the AR-LSAT test set (Zhong et al., 2022). In this work, we use many-sorted first-order logic, which is one of the variants of the standard first-order logic, to better reflect the scenarios of the real-world logical reasoning problems.

### A.1  SYNTAX

The syntax of many-sorted first-order logic is built on signatures. We first define signatures, and on top of that, we define variables, terms, atoms, literals, clauses, and formulas.

**Definition 5** (Signatures). A *signature* $\Sigma = (S, F, P)$ consists of countable sets of *sorts* $S$, *function symbols* $F$, and *predicate symbols* $P$. Each function symbol $f$ is associated with a type $s_1 \times \cdots \times s_n \to s$, where $n \geq 0$ and $s_1, \cdots, s_n, s \in S$. Function symbols with $n = 0$ (i.e. zero *arity*) are called *constants* of sort $s$. Each predicate symbol $p$ is associated with a type $s_1 \times \cdots \times s_n$, where $n \geq 1$ and $s_1, \cdots, s_n \in S$.

Let $\Sigma$ be a signature. A set $X$ of $\Sigma$-*variables* (or simply *variables*) is a countable set of variable names. Each variable name is associated with a sort in $\Sigma$. Based on variables, we define terms. Intuitively, terms are variables or functions applied to a tuple of other terms.

**Definition 6** (Terms). $\Sigma$-*terms* over $X$ (or simply *terms*) are defined as follows. Each variable $x \in X$ of sort $s$ is a term of sort $s$. If $t_1, \cdots, t_n$ are terms of sorts $s_1, \cdots, s_n$, respectively, and $f$ is a function symbol of type $s_1 \times \cdots \times s_n \to s$, then $f(t_1, \cdots, t_n)$ is a term of sort $s$.

Based on terms, we define atoms. Intuitively, atoms are predicates applied to a tuple of terms.

**Definition 7** (Atoms). $\Sigma$-*atoms* over $X$ (or simply *atoms*) are defined as follows. If $t_1, \cdots, t_n$ are $\Sigma$-terms over $X$ of sorts $s_1, \cdots, s_n$, respectively, and $p$ is a predicate symbol with the type $s_1 \times \cdots \times s_n$, then $p(t_1, \cdots, t_n)$ is an atom.

Additionally, *literals* are atoms or negation of atoms, and *clauses* are disjunctions of literals. We finally define formulas by using above definitions with logical connectives and quantifiers.

**Definition 8** (Formulas). $\Sigma$-*formulas* over $X$ (or simply *formulas*) are defined as follows. Each $\Sigma$-atom over $X$ is a formula. If $\alpha$ and $\beta$ are formulas, then so are $\neg\alpha$, $\alpha \wedge \beta$, and $\alpha \vee \beta$. If $x \in X$ is a variable of sort $s$ and $\alpha$ is a formula, then so are $\exists x\, \alpha$ and $\forall x\, \alpha$.

We give an example of the definitions of first-order logic syntax with the following formula in the AR-LSAT test set.

**Example 1.** A signature is given as

$$\Sigma_1 = (\{positions, potters\}, F, P)$$

where $F = \{displayed, Reigel, 1, 6\}$ such that *displayed* has the type *positions* $\to$ *potters*, *Reigel* is a constant of sort *potters*, and 1 and 6 are constants of sort *positions*, and $P = \{\approx\}$ such that $\approx$ is of the type *positions* $\times$ *positions*. We note that the equality symbol $\approx$ is always implicit from the context. If $p$ is a variable of sort *positions*, then $p$ and *displayed*$(p)$ are $\Sigma_1$-terms of sort *positions*. Then, *displayed*$(p) \approx Reigel$, $p \approx 1$, and $p \approx 6$ are $\Sigma_1$-atoms. Finally, the following is one example of a $\Sigma_1$-formula

$$(\forall p : positions)\, (displayed(p) \approx Reigel) \to (p \approx 1 \vee p \approx 6)$$

which is the first-order logic translation of the natural language sentence "*Reigel's bowl can be displayed only in either position 1 or position 6*".

### A.2  SEMANTICS

The semantics of many-sorted first-order logic is indicated by structures. We first define structures and extend its concept to define interpretations. On top of those, we define theories with models and consequences.

A signature $\Sigma = (S, F, P)$ only describes the names of sorts, functions, and predicates. However, it does not describe assignments of elements to each sort and evaluations of functions and predicates on the elements of sorts. Structures add this information.

**Definition 9** (Structures). A $\Sigma$-*structure* $I$ assigns a non-empty domain set $D_s$ to each sort $s \in S$, a function $f^I : D_{s_1} \times \cdots \times D_{s_n} \to D_s$ for each function symbol $f \in F$ of type $s_1 \times \cdots \times s_n \to s$, and a predicate $p^I : D_{s_1} \times \cdots \times D_{s_n} \to \{F, T\}$ for each predicate symbol $p \in P$ of type $s_1 \times \cdots \times s_n$. Note that each constant $c$ of sort $s$ is mapped to an element $c^I \in D_s$.

To evaluate terms and formulas, we extend structures to variables and define interpretations.

**Definition 10** (Interpretations). $\Sigma$-*interpretation* $I$ over $X$ (or simply *interpretation*) is a $\Sigma$-structure that additionally assigns a value $x^I \in D_s$ to each variable $x \in X$ of sort $s$.

We denote $I \vDash \phi$ if an interpretation $I$ evaluates a formula $\phi$ to true. However, we are not usually interested in the evaluation of formulas in a given structure, but interested in specific meaning of functions and predicates. Theories deal with this problem.

**Definition 11** (Theories). A $\Sigma$-*theory* $T$ (or simply *theory*) is a non-empty set of $\Sigma$-structures.

To solve practical problems, we additionally define the followings. A $T$-*interpretation* is a $\Sigma$-interpretation $I$ that extends some structure in the theory $T$. A formula $\phi$ is $T$-*satisfiable* if $I \vDash \phi$ for some $T$-interpretation $I$. A formula $\phi$ is $T$-*valid*, denoted by $\vDash_T \phi$, if $I \vDash \phi$ for all $T$-interpretations $I$. We also introduce definitions to describe the relationship between an interpretation and a formula, and between two formulas.

**Definition 12** (Models). A $T$-interpretation $I$ such that $I \vDash \phi$ is called a $T$-*model* of $\phi$.

**Definition 13** (Consequences). A formula $\phi$ is a $T$-*consequence* of a formula $\psi$, denoted by $\psi \vDash_T \phi$, if $I \vDash \psi$ implies $I \vDash \phi$ for all $T$-interpretations $I$.

We give an example of the definitions of first-order logic semantics by continuing Example 1.

**Example 2.** Let us consider the extended signature $\Sigma_2 = (\{positions, potters\}, F, \{\approx\})$ where

$$F = \{displayed, Larsen, Mills, Neiman, Olivera, Park, Reigel, Serra, Vance, 1, 2, 3, 4, 5, 6\}.$$

There are many possible $\Sigma_2$-structures, and one exemplar structure $I_2$ is:

- $D_{\text{positions}} = \{1, 2, 3, 4, 5, 6\}$,

- $D_{\text{potters}} = \{Larsen, Mills, Neiman, Olivera, Park, Reigel, Serra, Vance\}$,

- $Larsen^{I_2} = Larsen, Mills^{I_2} = Mills, Neiman^{I_2} = Neiman, Olivera^{I_2} = Olivera,$ $Park^{I_2} = Park, Reigel^{I_2} = Reigel, Serra^{I_2} = Serra, Vance^{I_2} = Vance,$

- $1^{I_2} = 1, 2^{I_2} = 2, 3^{I_2} = 3, 4^{I_2} = 4, 5^{I_2} = 5, 6^{I_2} = 6$, and

- $displayed^{I_2}(1) = Reigel, displayed^{I_2}(2) = Larsen, displayed^{I_2}(3) = Reigel,$ $displayed^{I_2}(4) = Reigel, displayed^{I_2}(5) = Reigel, displayed^{I_2}(6) = Reigel.$

Since we do not consider any additional variable here, $\Sigma_2$-interpretation $I_2'$ is the same as $I_2$.

Now, let the theory $T_2$ be the $\Sigma_2$-theory consisting only of the $\Sigma_2$-structure $I$ which has the same domain assignment to each sort and the same constant function assignments with $I_2$. $I_2$ is a $T_2$-*interpretation*. If we consider the formula $\phi$ in Example 1, $\phi$ is $T_2$-*satisfiable*, but not $T_2$-*valid*.

Lastly, let us determine if $I_2$ satisfies $\phi$. Reigel's bowl is displayed in the positions 1 and 6, but it is also displayed in positions 3, 4, and 5. A $T_2$-*interpretation* $I_2$ does not satisfy the formula $\phi$, which means $I_2$ is not a $T_2$-*model* of $\phi$.

## B    FIRST-ORDER LOGIC COMPLEXITY

To quantitatively measure the complexity of first-order logic formulas, we use a parameter following Arias & Khardon (2003). The target formula is first transformed into an expression in a conjunctive normal form (CNF). Then, the complexity parameter is defined as a sum of three components, the number of clauses in the CNF expression, the maximum number of distinct terms in any clause of the CNF expression, and the maximum number of literals in any clause of the CNF expression. We implement the parameter by using a Z3 theorem prover (De Moura & Bjørner, 2008).

We present two exemplar formulas and their complexity from the AR-LSAT test set. The first example is as follows:

$$(\exists C : children)\,(\neg(C \approx Juan) \wedge (assigned(C) \approx assigned(Juan))).$$

The formula is already in the CNF expression. It contains two clauses; $\neg(C \approx Juan)$ and $assigned(C) \approx assigned(Juan)$. The first clause has two distinct terms; $C$ and *Juan*, and one literal; $\neg(C \approx Juan)$. The second clause has four distinct terms; $C$, *Juan*, $assigned(C)$, and $assigned(Juan)$, and one literal; $assigned(C) \approx assigned(Juan)$. The measured complexity is $2 + 4 + 1 = 7$.

The second example is as follows:

$$(onSale(newPop) \wedge onSale(usedPop)) \rightarrow (onSale(newSoul) \wedge onSale(usedSoul)).$$

The formula is first transformed into a CNF expression as follows:

$$(onSale(newSoul) \vee \neg onSale(newPop) \vee \neg onSale(usedPop)) \wedge$$

$$(onSale(usedSoul) \vee \neg onSale(newPop) \vee \neg onSale(usedPop)).$$

It contains two clauses. The first clause has three distinct terms; *newSoul*, *newPop*, and *usedPop*, and three literals; $onSale(newSoul)$, $\neg onSale(newPop)$, and $\neg onSale(usedPop)$. The second clause has the same number of distinct terms and literals. The measured complexity is $2 + 3 + 3 = 8$.

## C    COMPONENT ACCUMULATION RULES

We describe rules for component accumulation according to a logical dependency structure. It aims to add components on simple subsentences to compose more complex subsentences in a specific order that preserves the underlying first-order logic semantics. We note that the rules are deduced from the definition of logical dependency structure (Section 3.1). The followings are the rules for component accumulation:

Rule 1.  Start with copying logic units.

Rule 2.  If a logic dependent $D$ is the only dependent of a logic unit $U$, then integrate $D$ into $U$ and add the updated U.

Rule 3.  If a logic dependent $D_1$ depends on another logic dependent $D_2$, then integrate $D_1$ into a logic unit $U$ that includes $D_2$ and add the updated $U$.

Rule 4.  If more than one logic dependents $D_1, D_2, \cdots, D_k$ depend on a logic unit $U$, then add $k$ sentences that include $U$ and each logic dependent $D_i$ ($i = 1, 2, \cdots, k$). After that, integrate all logic dependents into $U$ and add the updated $U$.

Rule 5.  If more than one components $X_1, X_2, \cdots, X_k$ depend on a logic coupler $C$, then integrate all components into $C$ and add the updated $C$.

We present three examples of component accumulation given sentences and their logical dependency structures in the following section.

# D  EXAMPLES OF LOGICAL DEPENDENCY PARSING AND COMPONENT ACCUMULATION

We present three examples of logical dependency parsing and component accumulation. For component accumulation, the applied rules and corresponding logical dependencies are indicated for each accumulation step. (Integration) is additionally inserted for Rule 4 to maintain consistency.

---

**Input**

*Target sentence*:
Each story is assigned a team of two interns—one photographer's assistant and one writer's assistant.

---

*Declarations*:
$interns = \{Farber, Gombarick, Hall, Jackson, Kanze, Lha\}$
$stories = \{Romania, Spain, Tuscany\}$
$assistants = \{photographer, writer\}$
$assigned = Function(interns \rightarrow stories)$
$trained = Function(interns \rightarrow assistants)$

**Output**

*Logical dependency parsing*:
$U_1$="Each story is assigned an intern.", $U_2$="There is an intern—a photographer's assistant.",
$U_3$="There is an intern—a writer's assistant.", $C_1$="(merge)", $C_2$="(merge)", $C_3$="a team of two ... and"
$U_1 \rightarrow C_1$, $U_2 \rightarrow C_1$, $U_1 \rightarrow C_2$, $U_3 \rightarrow C_2$, $C_1 \rightarrow C_3$, $C_2 \rightarrow C_3$

---

*Component accumulation*:
Rule 1: Each story is assigned an intern. ($U_1$)
Rule 1: There is an intern—a photographer's assistant. ($U_2$)
Rule 1: There is an intern—a writer's assistant. ($U_3$)
Rule 5: Each story is assigned an intern—a photographer's assistant. ($U_1 \rightarrow C_1$, $U_2 \rightarrow C_1$)
Rule 5: Each story is assigned an intern—a writer's assistant. ($U_1 \rightarrow C_2$, $U_3 \rightarrow C_2$)
Rule 5: Each story is assigned a team of two interns—one photographer's assistant and one writer's assistant. ($C_1 \rightarrow C_3$, $C_2 \rightarrow C_3$)

---

**Input**

*Target sentence*:
Xena and at least three other technicians repair radios.

---

*Declarations*:
$technicians = \{Stacy, Urma, Wim, Xena, Yolanda, Zane\}$
$machines = \{radios, televisions, VCRs\}$
$repairs = Function(technicians, machines \rightarrow bool)$

**Output**

*Logical dependency parsing*:
$U_1$="Xena repairs radios.", $D_1$="and three other technicians", $D_2$="exactly"
$D_1 \rightarrow U_1$, $D_2 \rightarrow D_1$

---

*Component accumulation*:
Rule 1: Xena repairs radios. ($U_1$)
Rule 2: Xena and three other technicians repair radios. ($D_1 \rightarrow U_1$)
Rule 3: Xena and at least three other technicians repair radios. ($D_2 \rightarrow D_1$)

Figure 5: Example of logical dependency parsing and component accumulation on AR-LSAT.

---

**Input**

*Target sentence*:
Each candidate must speak either first or second at at least one of the meetings.

- - - - - - - - - - - - - - - - - - - - - - - - - - - - - - - - - - - - - - - - - - - - - - - - - - - - - - - - - - - - - - - - - - - - - - - - - - - - - - - - - -

*Declarations*:
$candidates = \{Q, R, S, T, U\}$
$meetings = \{1, 2, 3\}$
$orders = \{1, 2, 3, 4, 5\}$
$speaks = Function(candidates, meetings \rightarrow orders)$

---

**Output**

*Logical dependency structure*:
$U_1$="Each candidate speaks at one of the orders at one of the meetings.", $D_1$="must", $D_2$="at least",
$D_3$="either first or second"
$D_1 \rightarrow U_1, D_2 \rightarrow U_1, D_3 \rightarrow U_1$

- - - - - - - - - - - - - - - - - - - - - - - - - - - - - - - - - - - - - - - - - - - - - - - - - - - - - - - - - - - - - - - - - - - - - - - - - - - - - - - - - -

*Component accumulation*:
Rule 1: Each candidate speaks at one of the orders at one of the meetings. $(U_1)$
Rule 4: Each candidate must speak at one of the orders at one of the meetings. $(D_1 \rightarrow U_1)$
Rule 4: Each candidate speaks at one of the orders at at least one of the meetings. $(D_2 \rightarrow U_1)$
Rule 4: Each candidate speaks either first or second at one of the meetings. $(D_3 \rightarrow U_1)$
Rule 4: Each candidate must speak either first or second at at least one of the meetings. (Integration)

---

Figure 6: Another example of logical dependency parsing and component accumulation on AR-LSAT.

## E  SAT Solver Function Prediction on AR-LSAT

For the AR-LSAT dataset, we additionally need to predict a solver function $f_{\text{solver}}$ according to the question of the logical reasoning problem. For instance, if the question is "Which of the queries CAN be true?", then we need to assign a function that checks a *satisfiability* of the query given the constraints. If the question is "Which of the queries MUST be true?", then we need to assign a function that checks a *validity* of the query given the constraints.

Logic-LM predicts a solver function together with the first-order logic translation by a single inference as follows:

$$\hat{\mathcal{T}}, \{\hat{\varphi}_k, \hat{NL}(\varphi_k)\}_{k=1}^{K+1}, \hat{f}_{\text{solver}} \sim P_{\text{LLM}}(\mathcal{T}, \{\varphi_k, NL(\varphi_k)\}_{k=1}^{K+1}, f_{\text{solver}} \mid x, \mathbf{x}_{\text{fs}}). \tag{6}$$

For CLOVER, to incorporate solver function prediction in our problem formulation in Eq. 2, we perform this prediction at the preprocessing step as follows:

$$\hat{\mathcal{T}}, \{\hat{NL}(\varphi_k)\}_{k=1}^{K+1}, \hat{f}_{\text{solver}} \sim P_{\text{LLM}}(\mathcal{T}, \{NL(\varphi_k)\}_{k=1}^{K+1}, f_{\text{solver}} \mid x, \mathbf{x}_{\text{fs}}^{\text{prep}})$$
$$\hat{\varphi}_k = \text{CLOVER}(\hat{\mathcal{T}}, \hat{NL}(\varphi_k), \mathbf{X}_{\text{fs}}), \forall k \in \{1, 2, \cdots, K+1\}. \tag{7}$$

# F  DATASET STATISTICS

In this section, we present dataset statistics of the logical reasoning tasks in Table 4. We describe the details in the following paragraphs. If not mentioned, we use the entire test set provided by the dataset.

**AR-LSAT-annotated.**  We annotate a representative subset of AR-LSAT test set (Zhong et al., 2022) to measure first-order logic translation accuracy at a formula-level. First, we sample the first problem from each set of problems that share the same context in the AR-LSAT test set. Then, we preprocess the logical reasoning problem to generate a theory $\hat{\mathcal{T}}$ and a set of natural language sentences, following the steps in Section 2. For each problem, we note that the sentences for constraints represent diverse first-order logic semantics while the sentences for five queries represent (nearly) the same first-order logic semantics. We therefore exclude other four queries and leave only the first one. For each sentence, we carefully annotate $\hat{\mathcal{T}}$-*satisfiable* first-order logic formula and double-check its correctness. If a $\hat{\mathcal{T}}$ cannot express the context of the logical reasoning problem, we exclude the sentences in that problem. As a result, we collect a total of 305 annotated formulas.

**ZebraLogic.**  The ZebraLogic test test (Lin et al., 2025) consists of 1,000 zebra puzzles where the puzzle size varies from $2 \times 2$ to $6 \times 6$. There are 25 different puzzle sizes, and each size has 40 samples. To evaluate the models on the most challenging puzzles, we use six hardest puzzle sizes ($4 \times 6, 5 \times 5, 5 \times 6, 6 \times 4, 6 \times 5$, and $6 \times 6$) for our test set, which yields a total of 240 puzzles.

**Puzzle.**  The entire dataset (Srivastava et al., 2022) consists of 1,000 samples. To split a test set, we sample the last 200 samples in the order of the samples listed in the dataset.

**Symbol.**  The entire dataset (Srivastava et al., 2022) includes 990 samples, which are categorized into five subsets (plain, adversarial, tricky, agnostic name-side, and agnostic emoji-side) with the same size. All examples in different subsets share the same logical meaning with each other where the only difference is the semantic link between the emojis and their names. To focus on a first-order logic translation, we evaluate the models on the plain subset which includes 198 samples. The plain subset consists of three subgroups with the same size categorized by their difficulties. To construct a test set, we sample the second half of each subgroup in the order of the samples listed in the dataset, which yields a total of 99 samples.

**Deduction.**  The entire dataset (Srivastava et al., 2022) consists of 1,500 samples. We use the test set following Pan et al. (2023) which consists of 300 samples.

**ProofWriter.**  We use the test set following Pan et al. (2023), which is a set of randomly sampled 600 examples from the most challenging depth-5 subset.

Table 4: Number of few-shot examples, test examples, and options, and license of the logical reasoning tasks used in the paper.

| Dataset | # Shot | # Test | # Options | License |
|---|---|---|---|---|
| AR-LSAT (Zhong et al., 2022) | 5 | 231 | 5 | MIT license |
| AR-LSAT-annotated | N/A | 305 | N/A | MIT license |
| ZebraLogic (Lin et al., 2025) | 1 | 240 | N/A | Apache 2.0 |
| Puzzle (Srivastava et al., 2022) | 1 | 200 | 2,3,4,5 | Apache 2.0 |
| Symbol (Srivastava et al., 2022) | 1 | 99 | 5 | Apache 2.0 |
| Deduction (Srivastava et al., 2022) | 2 | 300 | 3,5,7 | Apache 2.0 |
| FOLIO (Han et al., 2022) | 2 | 203 | 3 | CC-BY-SA-4.0 license |
| ProofWriter (Tafjord et al., 2021) | 1 | 600 | 3 | CC BY 4.0 |

## G  PERFORMANCE ON DIFFERENT LANGUAGE MODELS

Table 5 compares the performance of CLOVER and the neurosymbolic approach baselines on different languange models. We include three additional language models including `gpt-4o-mini`, `gpt-3.5-turbo`, and `gpt-3.5-turbo-instruct`, which the first two are chat-focused models and the other one is an instruction-following model. We evaluate these models on the Puzzle and Symbol datasets. If the symbolic solver cannot execute the solution, then we take random guesses. We exclude the performance of SymbCoT using `gpt-3.5-turbo-instruct` since the prompt including few-shot examples exceeds the context window of the language model. The results show that CLOVER clearly outperforms the baselines across different language models.

Table 5: Performance with different language models using CLOVER and neurosymbolic approach baselines.

| | Puzzle | | | Symbol | | |
|---|---|---|---|---|---|---|
| | Logic-LM | SymbCoT | CLOVER | Logic-LM | SymbCoT | CLOVER |
| *gpt-4o-mini* | 42.5 | 60.0 | **60.5** | 38.4 | 46.5 | **71.7** |
| *gpt-3.5-turbo* | 42.5 | 35.0 | **63.5** | 24.2 | 27.3 | **60.6** |
| *gpt-3.5-turbo-instruct* | 46.0 | N/A | **59.0** | 50.5 | N/A | **70.7** |

## H  INFERENCE TIME COSTS

It is difficult to measure inference time costs for methods that use LLMs with API calls. Specifically, inference time significantly depends on the current network traffic of an API, and the number of parameters are unknown. Despite this limitation, we compare CLOVER and the baselines by their API usage costs, which is a reliable way to measure inference time costs.

For comparison, we use `gpt-4o-mini` as a language model and measure the costs on the AR-LSAT annotated subset. We report the results in Table 6. CLOVER requires larger amount of inference costs compared to the baselines since the compositional first-order logic translation generates formulas for each logical dependency structure of a target sentence. However, the increased inference time cost is worth for the significant performance gain in Table 1.

Table 6:  Comparison of inference time costs using CLOVER and the baselines with `gpt-4o-mini`.

| | Costs (USD) |
|---|---|
| Standard | 0.02 |
| CoT | 0.02 |
| Logic-LM | 0.08 |
| SymbCoT | 0.15 |
| CLOVER | 0.30 |

# I    IMPACT OF SAT-BASED FIRST-ORDER LOGIC VERIFICATION

To further analyze an impact of using satisfiability in the verification algorithms, we compare those to two baselines: syntax consistency and LLM with instruction. First, we select the most frequent syntactically same formulas, which we call syntax consistency. Compared to logical consistency, logically equivalent but syntactically different formulas count as different formulas here. Second, we prompt LLM to select the most probable formula with reasoning, which we call LLM with instruction. LLM-based first-order logic verification is inspired by the previous works (Pan et al., 2023; Kirtania et al., 2024; Xu et al., 2024; Chen et al., 2024; Madaan et al., 2024; Ni et al., 2023).

We report the results in Table 7. The baselines show poor performance than the proposed SAT-based verification algorithms. Compared to a random selection, syntax consistency show 4.7% performance increment on AR-LSAT, but 0.8% marginal increment on ZebraLogic. LLM with instruction does not show any performance improvement on both tasks, which points out the limited capability of LLM to verify first-order logic formulas. These results show that SAT-based first-order logic verification is the most appropriate algorithm that fully covers first-order logic semantics.

Table 7: Comparison of different first-order logic verification approaches on AR-LSAT and ZebraLogic.

| Verification | AR-LSAT | ZebraLogic |
|---|---|---|
| Random | 55.0 | 70.0 |
| Syntax consistency | 59.7 | 70.8 |
| LLM w/ instruction | 53.3 | 70.0 |
| Logical consistency | 61.9 | 74.2 |
| Disproving | **62.8** | **75.4** |

## J  FEW-SHOT PROMPT EXAMPLES

---

**Prompt for Logical Dependency Parsing**

### Definition
(Explain definition of logical dependency structures)
### Instruction
Given declarations and a sentence, generate different possible logical dependency structures of the sentence. When generating a logical dependency structure, first declare each component and indicate the dependency of one and another. The followings rules must be satisfied:
1. All components except for one should be dependent on another component.
2. Conjunctions could be included in logic units or logic dependents, while not being allocated as logic couplers.
3. No logic coupler can be a head of logic dependent.
4. If a logic coupler or a logic dependent includes sets of words that are not adjacent, then separate them with "...".
5. Any logic dependent cannot be a conjunction itself.

-------------------------------------------------------------------------------------------------

### Declarations
```
technicians = EnumSort([Stacy, Urma, Wim, Xena, Yolanda, Zane])
machines = EnumSort([radios, televisions, VCRs])
repairs = Function([technicians, machines] -> [bool])
```
### Sentence
Xena and exactly three other technicians repair radios
### Structures
### 1:
U1="Xena repairs radios", U2="exactly three technicians repair radios", D1="other", C1="and"
D1 → U2; U1 → C1; U2 → C1
### 2:
U1="Xena repairs radios", D1="and three other technicians", D2="exactly"
D1 → U1; D2 → D1
### 3:
U1="Some technicians repair radios", D1="Xena and exactly three other technicians"
D1 → U1

-------------------------------------------------------------------------------------------------

### Declarations
```
technicians = EnumSort([Stacy, Urma, Wim, Xena, Yolanda, Zane])
machines = EnumSort([radios, televisions, VCRs])
repairs = Function([technicians, machines] -> [bool])
```
### Sentence
Stacy does not repair any type of machine that Yolanda repairs
### Structures
### 1:
U1="Stacy repairs any type of machine", U2="Yolanda repairs any type of machine", D1="does not", C1="that"
D1 → U1; U1 → C1; U2 → C1
### 2:
U1="Stacy repairs any type of machine", D1="does not", D2="that Yolanda repairs"
D1 → U1; D2 → U1

-------------------------------------------------------------------------------------------------

. . .

Figure 7: Prompt used for logical dependency parsing on AR-LSAT.

**Prompt for Component Accumulation**

### Definition
(Explain definition of logical dependency structures)
### Instruction
Given a sentence and its logical dependency structure, accumulate each component of the logical dependency structure to finally reach the original sentence. The followings are rules for accumulation:
1. Start with copying logic units.
2. If a logic dependent D is the only dependent of a logic unit U, then integrate D into U and accumulate the updated U.
3. If a logic dependent D1 depends on another logic dependent D2, then integrate D1 into a logic unit U that includes D2 and accumulate the updated U.
4. If more than one logic dependents D1, D2, ... Dk depend on a logic unit U, then accumulate k sentences that include U and each logic dependent Di (i=1, 2, ..., k). After that, integrate all logic dependents into U and accumulate the updated U.
5. If more than one components X1, X2, ... Xk depend on a logic coupler C, then integrate all components into C and accumulate the updated C.

----------------------------------------------------------------------------------

### Sentence
Xena and exactly three other technicians repair radios
### Structure
U1="Xena repairs radios", U2="exactly three technicians repair radios", D1="other", C1="and"
D1 → U2; U1 → C1; U2 → C1
### Accumulation
Xena repairs radios
exactly three technicians repair radios
exactly three other technicians repair radios
Xena and exactly three other technicians repair radios

----------------------------------------------------------------------------------

### Sentence
Stacy does not repair any type of machine that Yolanda repairs
### Structure
U1="Stacy repairs any type of machine", U2="Yolanda repairs any type of machine", D1="does not", C1="that"
D1 → U1; U1 → C1; U2 → C1
### Accumulation
Stacy repairs any type of machine
Yolanda repairs any type of machine
Stacy does not repair any type of machine
Stacy does not repair any type of machine that Yolanda repairs

----------------------------------------------------------------------------------

### Sentence
each candidate must speak either first or second at at least one of the meetings
### Structure
U1="each candidate must speak first at one of the meetings", U2="each candidate must speak second at one of the meetings", D1="at least", C1="either ... or"
D1 → U1; D1 → U2; U1 → C1; U2 → C1
### Accumulation
each candidate must speak first at one of the meetings
each candidate must speak second at one of the meetings
each candidate must speak first at at least one of the meetings
each candidate must speak second at at least one of the meetings
each candidate must speak either first or second at at least one of the meetings

----------------------------------------------------------------------------------
. . .

Figure 8: Prompt used for component accumulation on AR-LSAT.

**Prompt for Sequential Translation**

### Instruction
Given declarations, the task is to translate a sentence into a first order logic program. In order to do that, translate the given accumulation of components step by step to finally translate the original sentence.

------------------------------------------------------------------------------------------------

### Declarations
```
technicians = EnumSort([Stacy, Urma, Wim, Xena, Yolanda, Zane])
machines = EnumSort([radios, televisions, VCRs])
repairs = Function([technicians, machines] -> [bool])
```
### Sentence
Stacy does not repair any type of machine that Yolanda repairs
### Accumulation
Stacy repairs any type of machine
Stacy does not repair any type of machine
Stacy does not repair any type of machine that Yolanda repairs
### Translation
```
ForAll([m:machines], repairs(Stacy, m))
ForAll([m:machines], Not(repairs(Stacy, m)))
ForAll([m:machines], Implies(repairs(Yolanda, m), Not(repairs(Stacy,
m))))
```

------------------------------------------------------------------------------------------------

### Declarations
```
people = EnumSort([Vladimir, Wendy])
meals = EnumSort([breakfast, lunch, dinner, snack])
foods = EnumSort([fish, hot_cakes, macaroni, omelet, poached_eggs])
eats = Function([people, meals] -> [foods])
```
### Sentence
At no meal does Vladimir eat the same kind of food as Wendy
### Accumulation
At no meal does Vladimir eat the same kind of food as Wendy
### Translation
```
ForAll([m:meals], eats(Vladimir, m) != eats(Wendy, m))
```

------------------------------------------------------------------------------------------------

### Declarations
```
candidates = EnumSort([Q, R, S, T, U])
meetings = EnumSort([1, 2, 3])
speaks = Function([meetings, candidates] -> [int])
ForAll([m:meetings, c:candidates], And(1 <= speaks(m, c), speaks(m, c) <=
5))
```
### Sentence
R speaks fourth and fifth at meeting 1
### Accumulation
R speaks fourth at meeting 1
R speaks fifth at meeting 1
R speaks fourth and fifth at meeting 1
### Translation
```
speaks(1, R) == 4
speaks(1, R) == 5
And(speaks(1, R) == 4, speaks(1, R) == 5)
```

------------------------------------------------------------------------------------------------

...

Figure 9: Prompt used for sequential translation on AR-LSAT.

---

**Prompt for Disproving by Counter-Interpretation**

### Instruction
Your task is to determine whether the given solution can be included in any of the possible scenarios that arise when the given sentence is set as a condition. First, you need to calculate all possible scenarios that can occur under the given sentence as a condition, and then check if the situation obtained by interpreting the solution through the given declarations falls into one of those scenarios. As shown in the various examples below, you should provide your answer along with a logical explanation for the reasoning behind it. Keep in mind that you should focus on logical conditions rather than relying on commonsense when performing the task.

-------------------------------------------------------------------------------------------

### Input
Declarations:
```
people = EnumSort([Vladimir, Wendy])
meals = EnumSort([breakfast, lunch, dinner, snack])
foods = EnumSort([fish, hot_cakes, macaroni, omelet, poached_eggs])
eats = Function([people, meals] -> [foods])
```
Solution:
```
eats: Vladimir, breakfast -> fish
eats: Vladimir, lunch -> poached_eggs
eats: Vladimir, dinner -> macaroni
eats: Vladimir, snack -> hot_cakes
eats: Wendy, breakfast -> fish
eats: Wendy, lunch -> poached_eggs
eats: Wendy, dinner -> omelet
eats: Wendy, snack -> fish
```
Sentence:
Neither of them eats the same kind of food more than once during the day
### Response
Analysis:
Vladimir eats fish, poached eggs, macaroni, and hot cakes on a single day. Vladimir does not eat the same kind of food more than once during the day. Wendy eats fish, poached eggs, omelet, and fish on a single day. Wendy eats fish more than once during the day. Therefore, the solution does not logically satisfy the sentence, "Neither of them eats the same kind of food more than once during the day".
Answer:
No

-------------------------------------------------------------------------------------------

### Input
Declarations:
```
candidates = EnumSort([Q, R, S, T, U])
meetings = EnumSort([1, 2, 3])
speaks = Function([meetings, candidates] -> [int])
ForAll([m:meetings, c:candidates], And(1 <= speaks(m, c), speaks(m, c) <= 5))
```
Solution:
```
speaks: 1, Q -> 1
speaks: 1, R -> 3
speaks: 1, S -> 2
speaks: 1, T -> 4
speaks: 1, U -> 5
...
```
Sentence:
no two candidates will speak in the same time slot as each other at any meeting
### Response
Analysis:
At meeting 1, each candidate speaks 1st, 3rd, 2nd, 4th, and 5th. No two candidates speak in the same time slot as each other at meeting 1. At meeting 2, each candidate speaks 2nd, 5th, 3rd, 4th, and 1st. No two candidates speak in the same time slot as each other at meeting 2. At meeting 3, each candidate speaks 5th, 1st, 2nd, 4th, and 3rd. No two candidates speak in the same time slot as each other at meeting 3. Therefore, the solution logically satisfies the sentence, "no two candidates will speak in the same time slot as each other at any meeting".
Answer:
Yes

-------------------------------------------------------------------------------------------
. . .

Figure 10: Prompt used for disproving by counter-interpretation on AR-LSAT.

## K EXTENSIVE ERROR ANALYSIS OF LOGIC-LM

We present in-depth error analysis of Logic-LM (Pan et al., 2023) for first-order logic translation and the corresponding results of CLOVER on the AR-LSAT annotated subset. We follow Pan et al. (2023) and Ye et al. (2024) for the specification of declarations and formulas of first-order logic. We colorize incorrect translations as red and describe the reason in the following.

---

**Input**

*Target sentence*:
Each locker must be assigned to either one or two children, and each child must be assigned to exactly one locker.

---

*Declarations*:
```
children = EnumSort([Fred, Juan, Marc, Paul, Nita, Rachel, Trisha])
lockers = EnumSort([1, 2, 3, 4, 5])
assigned = Function([children] -> [lockers])
```

---

**Output**

CLOVER:
```
And(ForAll([l:lockers], Or(Count([c:children], assigned(c) == l) ==
1, Count([c:children], assigned(c) == l) == 2)), ForAll([c:children],
Exists([l:lockers], assigned(c) == l)))
```

---

Logic-LM:
```
ForAll([l:lockers], Or(Count([c:children], assigned(c) == l) == 1,
And(Count([c:children], assigned(c) == l) == 2, Exists([b:boys, g:girls],
And(assigned(b) == l, assigned(g) == l)))))
```

---

**Error Analysis of Logic-LM**

It contains an additional constraint that the locker assigned to two children should contain one boy and one girl.

---

Figure 11: Example #1 of error analysis of Logic-LM.

---

**Input**

*Target sentence*:
Juan must share a locker.

---

*Declarations*:
```
children = EnumSort([Fred, Juan, Marc, Paul, Nita, Rachel, Trisha])
lockers = EnumSort([1, 2, 3, 4, 5])
assigned = Function([children] -> [lockers])
```

---

**Output**

CLOVER:
```
Exists([c:children], And(c != Juan, assigned(c) == assigned(Juan)))
```

---

Logic-LM:
```
Exists([c:children], assigned(Juan) == assigned(c))
```

---

**Error Analysis of Logic-LM**

The formula is always true. The variable c should not be equal to Juan.

---

Figure 12: Example #2 of error analysis of Logic-LM.

**Input**

*Target sentence*:
Nita's locker cannot be adjacent to Trisha's locker.

---

*Declarations*:
```
children = EnumSort([Fred, Juan, Marc, Paul, Nita, Rachel, Trisha])
lockers = EnumSort([1, 2, 3, 4, 5])
assigned = Function([children] -> [lockers])
```

**Output**

CLOVER:
```
ForAll([l1:lockers, l2:lockers], Implies(And(assigned(Nita) == l1,
assigned(Trisha) == l2), Abs(l1 - l2) != 1))
```

---

Logic-LM:
```
ForAll([l:lockers], Implies(assigned(Nita) == l, And(assigned(Trisha) !=
l, assigned(Trisha) != l+1, assigned(Trisha) != l-1)))
```

**Error Analysis of Logic-LM**

It means Nita's locker cannot be adjacent and the same to Trisha's locker. The additional constraint should be eliminated.

Figure 13: Example #3 of error analysis of Logic-LM.

**Input**

*Target sentence*:
Each division is toured at least once.

---

*Declarations*:
```
days = EnumSort([Monday, Tuesday, Wednesday, Thursday, Friday])
divisions = EnumSort([Operations, Production, Sales])
toured = Function([days] -> [divisions])
```

**Output**

CLOVER:
```
ForAll([d:divisions], Exists([day:days], toured(day) == d))
```

---

Logic-LM:
```
ForAll([d:divisions], Exists([d:days], toured(d) == d))
```

**Error Analysis of Logic-LM**

Variable d is simultaneously assigned to two sorts.

Figure 14: Example #4 of error analysis of Logic-LM.

| Input |
| --- |
| *Target sentence*: 
 Each building was owned by exactly one of the families. |
| *Declarations*: 
 `families = EnumSort([Trents, Williamses, Yandells])` 
 `buildings = EnumSort([forge, granary, inn, mill, stable])` 
 `owned = Function([families, buildings] -> [bool])` |

| Output |
| --- |
| CLOVER: 
 `ForAll([b:buildings], Exists([f:families], And(owned(f, b),` 
 `ForAll([f2:families], Implies(f2 != f, Not(owned(f2, b)))))))` |
| Logic-LM: 
 `ForAll([b:buildings], Exists([f:families], owned(f, b)))` |

| Error Analysis of Logic-LM |
| --- |
| It means each building was owned by at least one of the families, not exactly one of the families. |

Figure 15: Example #5 of error analysis of Logic-LM.

| Input |
| --- |
| *Target sentence*: 
 Neither the inn nor the mill belonged to the owner of the forge. |
| *Declarations*: 
 `families = EnumSort([Trents, Williamses, Yandells])` 
 `buildings = EnumSort([forge, granary, inn, mill, stable])` 
 `owned = Function([families, buildings] -> [bool])` |

| Output |
| --- |
| CLOVER: 
 `ForAll([f:families], Implies(owned(f, forge), And(Not(owned(f, inn)),` 
 `Not(owned(f, mill)))))` |
| Logic-LM: 
 `ForAll([b:buildings], Implies(owned(forge, b), Not(Or(owned(inn, b),` 
 `owned(mill, b)))))` |

| Error Analysis of Logic-LM |
| --- |
| The domain of the function does not match with the declarations. |

Figure 16: Example #6 of error analysis of Logic-LM.

**Input**

*Target sentence*:
Reigel's bowl can be displayed only in either position 1 or position 6.

---
*Declarations*:
```
potters = EnumSort([Larsen, Mills, Neiman, Olivera, Park, Reigel, Serra, Vance])
positions = EnumSort([1, 2, 3, 4, 5, 6])
displayed = Function([positions] -> [potters])
```

**Output**

CLOVER:
```
ForAll([p:positions], Implies(displayed(p) == Reigel, Or(p == 1, p == 6)))
```
---
Logic-LM:
```
Or(displayed(1) == Reigel, displayed(6) == Reigel)
```

**Error Analysis of Logic-LM**

It means Reigel's bowl is displayed in either position 1 or position 6. We note that Reigel's bowl cannot be displayed.

Figure 17: Example #7 of error analysis of Logic-LM.

**Input**

*Target sentence*:
If Nash is assigned to a committee, Nash must be the leader for that committee.

---
*Declarations*:
```
volunteers = EnumSort([Haddad, Joslin, Kwon, Molina, Nash])
committees = EnumSort([X, Y, Z])
positions = EnumSort([leader, secretary, treasurer])
assigned = Function([committees, positions] -> [volunteers])
```

**Output**

CLOVER:
```
ForAll([c:committees], Implies(Exists([p:positions], assigned(c, p) == Nash), assigned(c, leader) == Nash))
```
---
Logic-LM:
```
ForAll([c:committees], Implies(assigned(c, leader) == Nash, ForAll([p:positions], assigned(c, p) != Nash)))
```

**Error Analysis of Logic-LM**

The formula is always false. Variable p should not be leader.

Figure 18: Example #8 of error analysis of Logic-LM.

**Input**

*Target sentence*:
Nottingham and Sunnyside are both visited, but not on consecutive days.

--------------------------------------------------------------------------------------------------------------------

*Declarations*:
```
days = EnumSort([Monday, Tuesday, Wednesday, Thursday, Friday])
neighborhoods = EnumSort([Hidden_Hills, Lakeville, Nottingham, Oldtown,
Park_Plaza, Sunnyside])
visited = Function([days] -> [neighborhoods])
```

**Output**

CLOVER:
```
And(Exists([d:days], visited(d) == Nottingham), Exists([d:days],
visited(d) == Sunnyside), ForAll([d1:days, d2:days],
Implies(And(visited(d1) == Nottingham, visited(d2) == Sunnyside), Abs(d1 -
d2) > 1)))
```

--------------------------------------------------------------------------------------------------------------------

Logic-LM:
```
And(Exists([d:days], visited(d) == Nottingham), Exists([d:days],
visited(d) == Sunnyside), Not(Exists([d:days], And(visited(d) ==
Nottingham, visited(d+1) == Sunnyside))))
```

**Error Analysis of Logic-LM**

It means that Nottingham is visited the day before Sunnyside is visited. We note that the opposite should be also possible.

Figure 19: Example #9 of error analysis of Logic-LM.

