# OpenReview forum: "Divide and Translate: Compositional First-Order Logic Translation and Verification for Complex Logical Reasoning"
_ICLR.cc/2025/Conference — ICLR 2025 Poster_

### Official Review · Reviewer_W7Jx · 2024-10-28

**Soundness:** 2
**Presentation:** 2
**Contribution:** 3
**Rating:** 6
**Confidence:** 4

**Summary:**

Introduces a new algorithm, CLOVER, for solving logic questions in natural language, specifically by addressing the challenges in parsing natural language into the correct first-order logic so that an SAT solver can determine the answer.  To do this, the paper proposes translating the question into smaller pieces that accumulate based on how each logic unit in natural language relates to other logic units until the resulting sentence is the final sentence that needs to be tested.  Each accumulation from the previous step, including the final sentence, is translated into first-order logic, then the paper introduces two novel verification methods that check if the translations are valid and if there are contradictions in the final translation. The paper shows that with this accumulation strategy with their two verifiers, their method can outperform all baselines (including Logic-LM, a baseline that similarly translates sentences into FOL for SAT solvers) on common logic datasets like AR-LSAT, FOLIO, and ProofWriter.

**Strengths:**

- The results are promising; the authors do a fantastic job of motivating their new algorithm CLOVER by showing common failures of previous methods like Logic-LM, then show that their method fixes many of these errors (leading to the performance boost reported in Table 1).

- The method here is pretty novel. Breaking down sentences into atoms isn't too novel, but I haven't seen someone decode them all individually and progressively (tapping into the auto-regressive natural of LMs) to improve the performance of the translation.  The verification algorithms seem pretty intuitive (although they are described complexly), but again, despite being intuitive, I think they are fairly novel as well.

**Weaknesses:**

- The ablations in Table 3 need to be explained more clearly and then discussed.  What is "Is Clover?" and why is the simplest ablation (no clover, direct translation, no verification / essentially none of the new things introduced in this paper) outperforming Logic-LM on AR-LSAT by 10.9% already from Table 1? Does this mean that your direct translation prompt already improves over complex algorithms like Logic-LM? If so, this deflates the papers impact, so it should be addressed (it's possible I am missing something, but others will catch this too, so it's best to explain it away.)

- I believe the paper would benefit greatly from expanding on the models being evaluated; right now, only GPT-4o and GPT-4o-mini are evaluated.  Showing that CLOVER consistently outperforms the baseline methods across model classes would improve the impact of this work.

- (minor point) There is no discussion of inference-time compute costs for CLOVER vs. the other baselines.  I imagine the inference cost is significantly higher, but I am unsure how much.  Is this negligible compared to Logic-LM?  Is there a way to compare CLOVER with baselines that use the equivalent amount of compute during inference?  I think much of this point could be explained away with a textual justification (i.e., this isn't possible, or the compute costs are nearly equivalent, etc.), but I do think it should be mentioned.

- (minor point) Clarity in section 3 could be improved. I would use the example in Figure 2 to clearly define each variable mentioned in the text to help readers follow your algorithm.  For instance, defining with x^prep, T, phi_k, the mapping NL(phi), etc., with values from Figure 2 would help readers follow significantly. This could also be done in Figure 2 if you mark which parts of it are which variables.  The text gets very dense with variables that are derived from other variables quickly; having these concrete instantiations really helps.

**Questions:**

- I'm curious why the execution rate increases when using CLOVER.  As I read the methods section, it looked like CLOVER primarily helps with execution accuracy, but I didn't see much about how it would help repair/fix/generate better code for the SAT solver.

- It's reported that "CLOVER’s errors are primarily caused by preprocessing and other errors, which takes 78.6% of the total errors", do you have examples of this? Is this an error in the accumulation stage?  I think the paper does a great job of explaining where Logic-LM fails and why CLOVER is needed, but I think expanding on CLOVER errors is just as important to show where researchers can look next.

- How much of the performance gain seen in CLOVER is due to a higher execution rate (runnable code)? I think expanding on how the metrics in Table 2 are computed would be helpful. For example is `Execution Acc = (correct_and_executable_programs / all)` or `Execution Acc = (correct_and_executable_programs / executable_programs)`.  The latter, I think, helps distinguish if you are generating better executable problems or if you are only improving the execution rate (which maybe there is a simple fix to Logic-LM to help it create better executable problems)?

---

> ### Author Response · Authors · 2024-11-21
> **Rebuttal by Authors (1/3)**
>
> We thank the reviewer for the constructive and detailed comments. **Q** denotes a Question, and **W** denotes a Weakness not addressed in the Questions.
>
> **W1. In Table 3, what is "Is Clover?" and why is the simplest ablation outperforming Logic-LM on AR-LSAT?**
>
> A. To answer the first question, "Is Clover?" means if a method is the proposed CLOVER or if it is one of the ablations. In Table 3, the last two rows correspond to CLOVER, which use both compositional translation and verification. In contrast, the first three rows correspond to the ablations, which do not use either compositional translation (i.e., use direct translation) or verification or both. To clarify this, we add related captions in Table 3.
>
> To answer the second question, the result that the simplest ablation outperforming Logic-LM only applies for the AR-LSAT dataset, and it comes from the specialized data characteristics. Unlike other datasets, we additionally need to predict solver functions according to the question of the logical reasoning problem. For instance, if the question is "Which of the queries CAN be true?", then we need to assign a function that checks a *satisfiability* of the query given the constraints. If the question is "Which of the queries MUST be true?", then we need to assign a function that checks a *validity* of the query given the constraints.
> Logic-LM predicts a solver function together with the first-order logic translation by a single inference in Eq. 1. For CLOVER, to incorporate solver function prediction in our problem formulation in Eq. 2, we perform this prediction at the preprocessing step. Comparing those two solver predictions, the latter one has significantly less load to an LLM since the task gets much simpler. Therefore, using the latter one improves accuracy to predict solver functions, where 10.9% gain comes from this. To support this analysis, we compare performance of the simplest ablation and Logic-LM on the three other datasets in the following table.
> |    |  Logic-LM  |  simplest ablation  |
> |------|:------:|:------:|
> | ZebraLogic |  45.4  |  45.4  |
> | Puzzle     |  64.0  |  64.0  |
> | Symbol     |  81.8  |  80.8  |
>
> The results indicate that the simplest ablation and Logic-LM have almost the same performance on the other datasets. This is because the other datasets use a single solver function, where there is no need for solver function predictions. We add the details of the solver function prediction on AR-LSAT in Appendix E. Overall, the performance gain on AR-LSAT is a side effect of the problem formulation.
>
> **W2. I believe the paper would benefit greatly from expanding on the models being evaluated.**
>
> A. We include the answer to Q3 of the reviewer 5CZv in the following.
>
> > In this paper, we evaluate CLOVER on two language models, gpt-4o and gpt-4o-mini (The results using gpt-4o-mini are presented in Table 5, Appendix G). For further evaluation, we include additional two language models including gpt-3.5-turbo and gpt-3.5-turbo-instruct, which are another chat-focused model and an instruction-following model, respectively. As in Table 5, we compare performance of CLOVER and neurosymbolic approach baselines on the Puzzle and Symbol datasets. If the symbolic solver cannot execute the solution, then we take random guesses. We exclude the performance of SymbCoT using gpt-3.5-turbo-instruct since the prompt including few-shot examples exceeds the context window of the language model. The following results show that CLOVER clearly outperforms the baselines across different language models. We add this results in Appendix G.
> > Due to the limitation of computational resources, we mainly use OpenAI models, but we will expand our evaluation on other proprietary models and open-sourced models.
> >
> > Performance on Puzzle dataset using different language models.
> > |  Puzzle  |  Logic-LM  |  SymbCoT  |  CLOVER  |
> > |------|:------:|:------:|:------:|
> > | gpt-4o-mini            |  42.5  |  60.0  |  **60.5**  |
> > | gpt-3.5-turbo          |  42.5  |  35.0  |  **63.5**  |
> > | gpt-3.5-turbo-instruct |  46.0  |  N/A  |  **59.0**  |
> >
> > Performance on Symbol dataset using different language models.
> > |  Symbol  |  Logic-LM  |  SymbCoT  |  CLOVER  |
> > |------|:------:|:------:|:------:|
> > | gpt-4o-mini            |  38.4  |  46.5  |  **71.7**  |
> > | gpt-3.5-turbo          |  24.2  |  27.3  |  **60.6**  |
> > | gpt-3.5-turbo-instruct |  50.5  |  N/A  |  **70.7**  |

---

> > ### Author Response · Authors · 2024-11-21
> > **Rebuttal by Authors (2/3)**
> >
> > Due to the space limit, we continue our answering in the following.
> >
> > **W3. Can you discuss inference-time compute costs for CLOVER vs. the other baselines?**
> >
> > A. It is difficult to measure inference time costs for methods that use LLMs with API calls. Specifically, inference time significantly depends on the current network traffic of an API, and the number of parameters are unknown. Despite this limitation, we compare CLOVER and the baselines by their API usage costs, which is a reliable way to measure inference time costs.
> >
> > For comparison, we use gpt-4o-mini as a language model and measure the costs on the AR-LSAT annotated subset. Standard prompting and CoT prompting both cost 0.02 USD, Logic-LM costs 0.08 USD, SymbCoT costs 0.15 USD, and CLOVER costs 0.30 USD. CLOVER requires larger amount of inference costs compared to the baselines since the compositional first-order logic translation generates formulas for each logical dependency structure of a target sentence. However, we think that the increased inference time cost is worth for the significant performance gain in Table 1. We add this disscusion in Appendix H.
> >
> > **W4. Clarity in section 3 could be improved.**
> >
> > A. Following the reviewer's feedback, we add corresponding mathematical notations of Section 3 to Figure 2.
> >
> > **Q1. Why does the execution rate increase when using CLOVER?**
> >
> > A. We find out that direct translations of complex logical sentences using Logic-LM often include syntax errors. In contrast, CLOVER reduces these errors by initiating translation from atomic subsentences, which improves the execution rate.
> > For example, a theory includes declarations of two types of sorts, $\textit{families}$ and $\textit{buildings}$, a predicate named $\textit{owned}$ of type $\textit{families} \times \textit{buildings}$, and constants named $\textit{inn}$, $\textit{mill}$,  and $\textit{forge}$ of sort $\textit{buildings}$. The target sentence is "Neither the inn nor the mill belonged to the owner of the forge.". Logic-LM translates this sentence as $(\forall b : \textit{buildings})\, (\textit{owned}(\textit{forge}, b) \rightarrow \lnot(\textit{owned}(\textit{inn}, b) \lor \textit{owned}(\textit{mill}, b)))$. This formula has a syntax error since the predicates include mismatched sort inputs (i.e., $\textit{inn}$, $\textit{mill}$,  and $\textit{forge}$ are not constants of sort $\textit{families}$).
> > However, CLOVER first translates an atomic subsentence "All families are the owner of the forge." into $(\forall f : \textit{families})\, (\textit{owned}(f, \textit{forge}))$, sequentially translates other subsentences, and finally translates the target sentence into $(\forall f : \textit{families})\, (\textit{owned}(f, \textit{forge}) \rightarrow \lnot(\textit{owned}(f, \textit{inn}) \lor \textit{owned}(f, \textit{mill})))$, which is both syntactically and semantically correct.
> >
> > **Q2. Do you have examples for the CLOVER's errors?**
> >
> > A. Most of CLOVER's errors on the AR-LSAT annotated subset are caused by incorrect preprocessing (the first equation of Eq. 2), which consists of incorrect target sentence generation and incorrect theory estimation.
> > For example, an original logical reasoning problem includes the following sentence: "There are three open positions on the appellate court and six open positions on the trial court, but not all of them will be filled at this time.". During the preprocessing, an LLM mistakenly omits the subsentence "but not all of them will be filled at this time." and generates the target sentence as "There are three open positions on the appellate court and six open positions on the trial court.", which degrades the meaning of the original context.
> > In another example, an estimated theory includes declarations of three types of sorts, $\textit{speakers}$, $\textit{rooms}$, and $\textit{times}$, and a function named $\textit{speech}$ of type $\textit{speakers} \rightarrow \textit{rooms} \times \textit{times}$. Here, the function declaration is incorrect since a function in first-order logic can only return a single type of sort.
> > Instead of a single LLM inference for preprocessing, put much work on target sentence generation and theory estimation would greatly benefit the overall performance.

---

> > > ### Author Response · Authors · 2024-11-21
> > > **Rebuttal by Authors (3/3)**
> > >
> > > Due to the space limit, we continue our answering in the following.
> > >
> > > **Q3. How much of the performance gain seen in CLOVER is due to a higher execution rate? I think expanding on how the metrics in Table 2 are computed would be helpful. For example is `Execution Acc = (correct_and_executable_programs / all)` or `Execution Acc = (correct_and_executable_programs / executable_programs)`.**
> > >
> > > A. To answer the reviewer's confusion on execution accuracy first, the latter one is correct. To clarify the metrics in Table 2, `Program Acc = (correct_and_executable_programs / all)`, `Execution Rate = (executable_programs / all)`, and `Execution Acc = (correct_and_executable_programs / executable_programs)`. These are briefly described in lines 417-419.
> > > CLOVER improves syntactic and semantic correctness of translation which contributes to the increased execution rate and execution accuracy, and these two collaboratively contribute to the final performance gain.

---

> > > > ### Comment · Reviewer_W7Jx · 2024-11-25
> > > > **Thank you for the reply!**
> > > >
> > > > > For comparison, we use gpt-4o-mini as a language model and measure the costs on the AR-LSAT annotated subset. Standard prompting and CoT prompting both cost 0.02 USD, Logic-LM costs 0.08 USD, SymbCoT costs 0.15 USD, and CLOVER costs 0.30 USD. CLOVER requires larger amount of inference costs compared to the baselines since the compositional first-order logic translation generates formulas for each logical dependency structure of a target sentence. However, we think that the increased inference time cost is worth for the significant performance gain in Table 1. We add this disscusion in Appendix H.
> > > >
> > > > Could you change this to token counts? (API-costs change over time).  But otherwise, this sounds very nice, and I appreciate the authors looking into it. From the table and explanation given, it looks like CLOVER is 3.75 times more expensive than the leading baseline (Logic-LM). Is there a way to make this comparison more equal? I am not entirely familiar with how Logic-LM could be scaled, but it would be nice to show a fair comparison between the two baselines, which would strengthen CLOVER's results, in my opinion.
> > > >
> > > > >Most of CLOVER's errors on the AR-LSAT annotated subset are caused by incorrect preprocessing (the first equation of Eq. 2), ...
> > > >
> > > > Thanks for showing some of them. I believe an extensive error analysis in the appendix (similar to how Logic-LM was done) would be beneficial.  Specifically showing how CLOVER fails on these examples, where CLOVER fails to outperform Logic-LM or other baselines, etc.  I didn't see them mentioned in the paper (but I could have missed them).
> > > >
> > > > I am upgrading my score to a 6 in light of these clarifications and changes.

---

> > > > > ### Author Response · Authors · 2024-11-27
> > > > > **Official Comment by Authors**
> > > > >
> > > > > We truly appreciate your response and would like to provide a brief reply to your comments.
> > > > >
> > > > > **Q1. Could you change API-costs to token counts?**
> > > > >
> > > > > A. We would also include token counts in the further revision.
> > > > >
> > > > > **Q2. Is there a way to make a fair comparison between Logic-LM and CLOVER?**
> > > > >
> > > > > A. Regarding the fair comparison of Logic-LM and CLOVER, instead of equalizing inference time costs, we use the same set of few-shot examples (line 408-409). To be specific, we derive the few-shot examples of CLOVER from those of Logic-LM. Since Logic-LM is hard to scale, we think our approach would be one of the reasonable ways to compare those two.
> > > > >
> > > > > **Q3. An extensive error analysis in the appendix would be beneficial.**
> > > > >
> > > > > A. We would also add extensive error analysis of CLOVER including the ones we show above in the further revision.

---

### Official Review · Reviewer_yp79 · 2024-10-29

**Soundness:** 3
**Presentation:** 1
**Contribution:** 2
**Rating:** 6
**Confidence:** 3

**Summary:**

Authors propose a novel method of using LLMs to translate natural language descriptions into a set of first-order logical forms. This novel method decomposes this challenging task into two steps. The first step is to translate a long and complex sentence into a number of short sentences, the second step is to translate each short sentence into simple first-order logical forms and the connections between/among these short sentences into corresponding logical connectors. Experiments on seven benchmark datasets greatly outperform current SOTA level.

**Strengths:**

It is reasonable to improve the translation quality by decomposing a complex sentence into several shorter sentences. Using SAT solvers certainly improve the quality.

**Weaknesses:**

Not all natural language sentences can be translated to first-order logic forms. Authors did not discuss what sentences cannot be translated.

Authors use symbolic SAT solver in evaluating and selecting correct first-order logical forms.  This limits the method only for the case where SAT solvers work.

Theoretically, the meaning of natural language is not logical formula. This work is valued within fixed benchmark datasets.

The formalism of the paper is not easy to read.

**Questions:**

1. line 115: "To save computational cost, we compare each one of logically equivalent formulas". You probably mean to "compare each logically equivalent formula". How can this save computational cost?

2. Line 149: how to read this formula in natural language?

3. What is the output for the sentence "A barber shaves all who do not shave themselves."?

4. How are "Declarations" created?

5. How to decide a sentence not fit for your system? (or how to decide an unintended input sentence?)

---

> ### Author Response · Authors · 2024-11-21
> **Rebuttal by Authors (1/2)**
>
> We thank the reviewer for the constructive and detailed comments. **Q** denotes a Question, and **W** denotes a Weakness not addressed in the Questions.
>
> **Misunderstanding in Summary**
>
> We observe that there is a slight misunderstanding on the reviewer's summary in the following:
>
> > The first step is to translate a long and complex sentence into a number of short sentences, the second step is to translate each short sentence into simple first-order logical forms and the connections between/among these short sentences into corresponding logical connectors.
>
> To clarify our method, the proposed compositional first-order logic translation consists of three steps. The first step is to parse a target sentence into logical dependency structures, the second step is to progressively accumulate each component of the structure while preserving its logical dependency, and the last step is to sequentially translate starting from the atomic subsentence until the target sentence.
> Not only that, the proposed first-order logic verification algorithms are another novelty of this paper. To fully leverage the semantics of first-order logic, 1) we select the most frequent logically equivalent formulas (Logical Consistency), or 2) we disprove one of two formulas by judging if a counter-interpretation satisfies the target sentence. These are visually summarized in Figure 2.
>
> **W1. Authors did not discuss what sentences cannot be translated to first-order logic forms. This work is valued within fixed benchmark datasets where the meaning of natural language is a logical formula.**
>
> A. We would like to emphasize that logical reasoning, where the problems are formally represented as a first-order logic, is one major category of reasoning. [2] states that logical reasoning plays a central role in intelligent systems for problem-solving, decision-making, and critical thinking. This paper and other prior works [1, 2, 3, 4] focus on enhancing the logical reasoning ability of LLMs. The other reasoning problems such as reading comprehension are out of the scope of this line of works.
>
> **W2. Authors use a SAT solver in evaluating and selecting correct first-order logical forms. This limits the method only for the case where SAT solvers work.**
>
> A. As we state in the answer of W1, every logical reasoning problem could be translated into first-order logic, which forms a SAT problem consists of a theory, constraints, and a query. Then, a sound SAT solver could investigate if two formulas are logically equivalent, or find counter-interpretations of the two formulas which are then used for disproving.
>
> **W3. The formalism of the paper is not easy to read.**
>
> A. We describe our problem formulation and method with mathematical notations to precisely explain those. To make those easy to read, we add corresponding mathematical notations to Figure 2.

---

> > ### Comment · Reviewer_yp79 · 2024-11-21
> >
> > > a sound SAT solver could investigate if two formulas are logically equivalent, or find counter-interpretations of the two formulas which are then used for disproving.
> >
> > I cannot totally agree with such a description of the "soundness" of a SAT solver. Could you please add some reference papers?

---

> > > ### Comment · Reviewer_yp79 · 2024-11-21
> > >
> > > > Formula #1 and #2 (which are syntactically the same) are both filtered out
> > >
> > > According to which rules, are both Formula #1 and #2 filtered out?
> > >
> > > My opinion, both structures are not correctly translated.

---

> > > > ### Author Response · Authors · 2024-11-21
> > > > **Official Comment by Authors**
> > > >
> > > > If we translate the reviewer's sentence, "A barber shaves all who do not shave themselves.", the output formula is $(\forall p : \textit{people})\, (\lnot \textit{shaves}(p, p) \rightarrow \textit{shaves}(\textit{barber}, p))$. In the above answer, we make a slight mistake in the verification that the formula is $\hat{T}$-satisfiable since the sentence is satisfied for the interpretation that the barber shaves all people including itself, which the formula is not filtered out (line 310-311). We revise our answer accordingly.
> > > > In addition, for better understanding, since the sentence comes from the barber paradox, we also translate the original sentence of the barber paradox, "A barber shaves all those, and those only, who do not shave themselves.". The output formula is $(\forall p : \textit{people})\, (\lnot \textit{shaves}(p, p) \Leftrightarrow \textit{shaves}(\textit{barber}, p))$. In this case, since there is no satisfiable interpretation (i.e., $\hat{T}$-unsatisfiable), the formula is filtered out (line 310-311). To ensure these results, we verify the formulas using a SAT solver. If the reviewer has a different opinion about the translation results, we would like to kindly ask the reviwer's opinion.

---

> > > > > ### Comment · Reviewer_yp79 · 2024-11-21
> > > > >
> > > > > Here, you need to use the descriptor &iota; to denote those people, right?
> > > > >
> > > > > Is this translation written by human experts, or by your system?
> > > > >
> > > > > If a logical system contains inconsistent statements, this system will assert any statement as true. Why does the SAT output nothing, when the input has inconsistent statements?

---

> > > > > > ### Author Response · Authors · 2024-11-22
> > > > > > **Official Comment by Authors**
> > > > > >
> > > > > > **Q1. Here, you need to use the descriptor ι to denote those people, right?**
> > > > > >
> > > > > > A. Yes, the reviewer is correct. In the preprocessing step, to implement the declarations in z3, we instruct an LLM to assign an arbitrary number of those people. The estimated declarations through the preprocessing step are shown in the following.
> > > > > > ```
> > > > > > # Declarations
> > > > > > people = [barber, person_1, person_2, person_3]
> > > > > > shaves = Function([people, people] -> [bool])
> > > > > > ```
> > > > > > For simplicity, we omit this detail in our first response as follows:
> > > > > > > An LLM returns a theory $\hat{T}$ that involves the following declarations: A sort named $\textit{people}$, a predicate named $\textit{shaves}$ of the type $\textit{people} \times \textit{people}$, and a constant named $\textit{barber}$ of sort $\textit{people}$ (a constant is a function with zero arity).
> > > > > >
> > > > > > **Q2. Is this translation written by human experts, or by your system?**
> > > > > >
> > > > > > A. All the translations shown above are generated by our system. We use gpt-4o as a language model.
> > > > > >
> > > > > > **Q3. If a logical system contains inconsistent statements, this system will assert any statement as true. Why does the SAT output nothing, when the input has inconsistent statements?**
> > > > > >
> > > > > > A. The filtering in line 310-311 is a simple process for eliminating unsatisfiable formulas using a SAT solver. Since an inconsistent formula is unsatisfiable, it is filtered out during this process. For better understanding, we show a pseudocode of this process in the following.
> > > > > > ```
> > > > > > # Formula
> > > > > > f = ForAll([p:people], Not(shaves(p, p)) == shaves(barber, p))
> > > > > >
> > > > > > # Checking satisfiability
> > > > > > solver = Solver()
> > > > > > solver.add(f)
> > > > > > print(solver.check() == sat)
> > > > > > ```
> > > > > > Here, the code returns `False` which means the formula is unsatisfiable, so we filtered out this formula.

---

> > > > > > > ### Comment · Reviewer_yp79 · 2024-11-22
> > > > > > >
> > > > > > > I updated my grade. I am not sure whether all translations were made by the system, as I cannot find the code and the data.

---

> > > ### Author Response · Authors · 2024-11-21
> > > **Official Comment by Authors**
> > >
> > > We would like to refer SatLM [1], which covers logical reasoning problems by using a SAT solver. In Section 2 of this paper, there is a following description:
> > > > First, because the SAT solver is *sound* (i.e., any assignment it produces satisfies the formula), the solution is correct by construction. Thus, assuming that the parsing is correct and $\hat{\Phi}$ and $\hat{Q}$ match $\Phi$ and $Q$, we have a proof that the solution is indeed correct.
> > >
> > > Here, $\hat{\Phi}$ and $\hat{Q}$ refer to the estimated formulas of constraints and a query, and $\Phi$ and $Q$ refer to the ground truth formulas.
> > >
> > > ---
> > > [1] Satlm: Satisfiability-aided language models using declarative prompting. Ye et al., 2024.

---

> ### Author Response · Authors · 2024-11-21
> **Rebuttal by Authors (2/2)**
>
> Due to the space limit, we continue our answering in the following.
>
> **Q1. Line 115: "To save computational cost, we compare each one of logically equivalent formulas". You probably mean to "compare each logically equivalent formula". How can this save computational cost?**
>
> A. To clarify the meaning of the sentence in line 115, here is an explanation using an example in Figure 2. After compositional first-order logic translation, there are six candidate formulas. For disproving by counter-interpretation, we should compare all the possible 15 pairs of six formulas. To save computational cost, we first group logically equivalent formulas which in fact there are only two logically different formulas, and then we could compare a single pair of the two formulas.
>
> **Q2. Line 149: How to read this formula in natural language?**
>
> A. For intuitive explanation, Eq. 1 means that in prior neurosymbolic approaches [1, 2, 3], an LLM translates a logical reasoning problem $x$ into a theory $\hat{T}$ and pairs of first-order logic formula and its natural language description.
> Note that a theory $\hat{T}$ includes 1) declarations of sorts, functions, and predicates and 2) the most commonly applied theories (e.g., theory of equality, arithmetic, etc.). For simplicity, we presume that a theory $\hat{T}$ always incorporates the most commonly applied theories.
>
> **Q3. What is the output for the sentence "A barber shaves all who do not shave themselves."?**
>
> A. Following our problem formulation, we first run a preprocessing step as in Eq. 2. An LLM returns a theory $\hat{T}$ that involves the following declarations: A sort named $\textit{people}$, a predicate named $\textit{shaves}$ of the type $\textit{people} \times \textit{people}$, and a constant named $\textit{barber}$ of sort $\textit{people}$ (a constant is a function with zero arity). An LLM also returns a target sentence same as the original sentence.
>
> Now, we perform compositional first-order logic translation to translate the target sentence under the estimated theory.
> 1. Logical Dependency Parsing
> Structure \#1:
> U1="A barber shaves all", D1="who do not shave themselves"
> D1 -> U1
> Structure \#2:
> U1="A barber shaves all", U2="People do not shave themselves", C1="(merge)"
> U1 -> C1, U2 -> C1
>
> 2. Component Accumulation
> For Structure \#1:
> 1\) A barber shaves all.
> 2\) A barber shaves all who do not shave themselves.
> For Structure \#2:
> 1\) A barber shaves all.
> 2\) People do not shave themselves.
> 3\) A barber shaves all who do not shave themselves.
>
> 3. Sequential Translation
> For Structure \#1:
> 1\) $(\forall p : \textit{people})\, (\textit{shaves}(\textit{barber}, p))$
> 2\) $(\forall p : \textit{people})\, (\lnot \textit{shaves}(p, p) \rightarrow \textit{shaves}(\textit{barber}, p))$ (Formula \#1)
> For Structure \#2:
> 1\) $(\forall p : \textit{people})\, (\textit{shaves}(\textit{barber}, p))$
> 2\) $(\forall p : \textit{people})\, (\lnot \textit{shaves}(p, p))$
> 3\) $(\forall p : \textit{people})\, (\lnot \textit{shaves}(p, p) \rightarrow \textit{shaves}(\textit{barber}, p))$ (Formula \#2)
>
> Lastly, we perform first-order logic verification to select the most probable formula.
> As described in line 310-311, we first filter out $\hat{T}$-*unsatisfiable* formulas using a SAT solver. Since Formula \#1 and \#2 are both $\hat{T}$-*satisfiable*, those are not filtered out. Those two formulas are syntactically the same, so we do not have to further proceed the verification. The output formula is $(\forall p : \textit{people})\, (\lnot \textit{shaves}(p, p) \rightarrow \textit{shaves}(\textit{barber}, p))$.
>
> **Q4. How are "Declarations" created?**
>
> A. The first equation in Eq. 2 (line 170) shows how declarations are generated. According to the equation, an LLM takes a logical reasoning problem $x$ as a input and generates a theory $\hat{T}$ and a set of target natural language sentences. A theory $\hat{T}$ corresponds to the "Declarations".
>
> **Q5. How to decide a sentence not fit for your system? (or how to decide an unintended input sentence?)**
>
> A. The first equation in Eq. 2 (line 170) also decides which sentence to translate. As the reviewer points out, a logical reasoning problem often contains sentences for declarations or even no information to solve the problem. To resolve this issue, we add a preprocessing step before applying CLOVER. Other neurosymbolic approaches [1, 2] decide which sentence to translate as in Eq. 1.
>
> ---
> [1] Satlm: Satisfiability-aided language models using declarative prompting. Ye et al., 2024.
> [2] Logic-lm: Empowering large language models with symbolic solvers for faithful logical reasoning. Pan et al., 2023.
> [3] Linc: A neurosymbolic approach for logical reasoning by combining language models with first-order logic provers. Olausson et al., 2023.
> [4] Faithful logical reasoning via symbolic chain-of-thought. Xu et al., 2024.

---

### Official Review · Reviewer_5CZv · 2024-11-04

**Soundness:** 3
**Presentation:** 2
**Contribution:** 2
**Rating:** 5
**Confidence:** 4

**Summary:**

The paper introduces CLOVER, an approach designed to enhance the translation of natural language logical problems into logical code, thereby improving the performance of language models on logical reasoning benchmarks. CLOVER achieves this by compositional translation of natural language into first-order logic and verification of logical semantics. The method involves parsing natural language sentences into logical dependency structures, translating these into first-order logic formulas, and employing verification algorithms to ensure accuracy. The authors demonstrate CLOVER's effectiveness on seven logical reasoning benchmarks, showing it outperforms previous neurosymbolic approaches.

**Strengths:**

- The paper presents a new approach by breaking down compositional translation into smaller steps and combining it with verification for logical reasoning tasks, leading to improved performance in neurosymbolic translation.
- The experimental results are robust using GPT4-o, showing improvements over other methods across multiple benchmarks.
- The authors propose two SAT-based first-order logic verification algorithms for selecting a sample from LLMs' logical code generations.

**Weaknesses:**

- The approach is primarily applicable to problems that can be represented in a SAT solver, limiting its generalizability to other reasoning datasets, such as those involving mathematical equations or visual components, e.g., MATH dataset.
- The core idea of breaking down tasks into subtasks and using multiple samples and tests (e.g., verification, self-reflection, deterministic tests) to select the best generation is not novel.
- The paper lacks comparison with chain-of-thought (CoT) based methods designed to improve implicit reasoning of language models, as in "reliable reasoning beyond natural language". These methods help the model extract information that is implied but not directly stated by interleaving natural language comments with logical code, and can alleviate the translation bottlenecks identified.
- The paper only reports results using one language model, making it unclear if the method would improve performance across different models and weakening the experimental results.

**Questions:**

1. Is it possible to extend CLOVER to improve performance on tasks that involve reasoning with data formats beyond natural language, such as mathematical equations or visual reasoning tasks?
2. Can the authors provide more insights into how CLOVER compares with CoT-based methods designed for improving implicit reasoning of LLMs?
3. Why not test CLOVER on a wider range of language models to assess its generalizability?

---

> ### Author Response · Authors · 2024-11-21
> **Rebuttal by Authors (1/2)**
>
> We thank the reviewer for the constructive and detailed comments. **Q** denotes a Question, and **W** denotes a Weakness not addressed in the Questions.
>
> **W1. The core idea of breaking down tasks into subtasks, using multiple samples, and verifying to select the best generation is not novel.**
>
> A. We would like to say that breaking down tasks into smaller ones and selecting the best generation is not our novelty. The novelty of this paper comes from how we break down logical sentences while preserving underlying logical structures and how we select the best first-order logic formula by fully leveraging the first-order logic semantics. To achieve the former one, we newly define a semantic parsing method called logical dependency parsing, and to achieve the latter one, we proposes two verification algorithms using a SAT solver. These contributions are summarized in the last paragraph of Introduction.
>
> **Q1. The approach is primarily applicable to problems that can be represented in a SAT solver, limiting its generalizability to other reasoning datasets, such as mathematical equations or visual reasoning tasks. Is it possible to extend CLOVER on these tasks?**
>
> A. We would like to emphasize that logical reasoning, where the problems are formally represented as a first-order logic, is one major category of reasoning. [2] states that logical reasoning plays a central role in intelligent systems for problem-solving, decision-making, and critical thinking. This paper and other prior works [1, 2, 3, 4] focus on enhancing the logical reasoning ability of LLMs. The other reasoning problems which require math or visual reasoning are out of the scope of this line of works.
>
> **Q2. The paper lacks comparison with CoT-based methods designed to improve implicit reasoning of language models [6]. Can the authors provide more insights into how CLOVER compares with these methods?**
>
> A. To clarify the meaning of CoT-based methods in the reviwer's question, we prepare two different answers.
> If the reviewer means CoT-based methods as extracting information (i.e., natural language comments with logical code) that is implied but not directly stated, SatLM [1], Logic-LM [2], and SymbCoT [4] belongs to the methods designed to improve implicit reasoning of language models as in [6]. Specfically, [1] and [2] shares nearly the same method with [6] where the main difference comes from what symbolic solvers they use. CoT [5] also extracts implicit information in natural language. We do compare CLOVER with these methods in Table 1, and the results show that CLOVER clearly outperforms all these methods. For CLOVER, the preprocessing step (first equation of Eq. 2) includes extracting natural language sentences that are implied but not directly stated.
>
> If the reviewer means CoT-based methods as CoT as is (i.e., step-by-step implicit reasoning through natural language), we want to emphasize the advantage of neurosymbolic approaches compared to CoT. While CoT falls short in complex logical reasoning tasks which need long sequence of reasoning, the neurosymbolic approaches including CLOVER resolve this issue by integrating a sound symbolic solver (line 39-48).

---

> ### Author Response · Authors · 2024-11-21
> **Rebuttal by Authors (2/2)**
>
> Due to the space limit, we continue our answering in the following.
>
> **Q3. The paper only reports results using one language model. Can the authors evaluate CLOVER on a wider range of language models?**
>
> A. In this paper, we evaluate CLOVER on two language models, gpt-4o and gpt-4o-mini (The results using gpt-4o-mini are presented in Table 5, Appendix G). For further evaluation, we include additional two language models including gpt-3.5-turbo and gpt-3.5-turbo-instruct, which are another chat-focused model and an instruction-following model, respectively. As in Table 5, we compare performance of CLOVER and neurosymbolic approach baselines on the Puzzle and Symbol datasets. If the symbolic solver cannot execute the solution, then we take random guesses. We exclude the performance of SymbCoT using gpt-3.5-turbo-instruct since the prompt including few-shot examples exceeds the context window of the language model. The following results show that CLOVER clearly outperforms the baselines across different language models. We add this results in Appendix G.
> Due to the limitation of computational resources, we mainly use OpenAI models, but we will expand our evaluation on other proprietary models and open-sourced models.
>
> Performance on Puzzle dataset using different language models.
> |  Puzzle  |  Logic-LM  |  SymbCoT  |  CLOVER  |
> |------|:------:|:------:|:------:|
> | gpt-4o-mini            |  42.5  |  60.0  |  **60.5**  |
> | gpt-3.5-turbo          |  42.5  |  35.0  |  **63.5**  |
> | gpt-3.5-turbo-instruct |  46.0  |  N/A  |  **59.0**  |
>
> Performance on Symbol dataset using different language models.
> |  Symbol  |  Logic-LM  |  SymbCoT  |  CLOVER  |
> |------|:------:|:------:|:------:|
> | gpt-4o-mini            |  38.4  |  46.5  |  **71.7**  |
> | gpt-3.5-turbo          |  24.2  |  27.3  |  **60.6**  |
> | gpt-3.5-turbo-instruct |  50.5  |  N/A  |  **70.7**  |
>
> ---
> [1] Satlm: Satisfiability-aided language models using declarative prompting. Ye et al., 2024.
> [2] Logic-lm: Empowering large language models with symbolic solvers for faithful logical reasoning. Pan et al., 2023.
> [3] Linc: A neurosymbolic approach for logical reasoning by combining language models with first-order logic provers. Olausson et al., 2023.
> [4] Faithful logical reasoning via symbolic chain-of-thought. Xu et al., 2024.
> [5] Chain-of-thought prompting elicits reasoning in large language models. Wei et al., 2022.
> [6] Reliable reasoning beyond natural language. Borazjanizadeh & Piantadosi, 2024.

---

> ### Author Response · Authors · 2024-11-28
> **Reminder for Paper Discussion**
>
> Dear Reviewer 5CZv,
>
> We appreciate the time and effort you have dedicated to reviewing our paper. We hope our responses and additional results have addressed your concerns. If you have any further questions or suggestions, we would be grateful to hear them. Thank you once again for your valuable feedback throughout this process!
>
> Best,
> Authors

---

> ### Author Response · Authors · 2024-12-03
> **Reminder for Paper Discussion**
>
> Dear Reviewer 5CZv,
>
> As the discussion period deadline approaches, we hope we have addressed your concerns regarding the novelty of our approach, its generalizability to other reasoning tasks, comparisons with CoT-based methods, and evaluations on additional language models.
> We would greatly appreciate it if you could provide a response to our rebuttal. Please let us know if you have any further questions or concerns!
>
> Best,
> Authors

---

### Meta-Review · Area_Chair_Xiaf · 2024-12-18

**Metareview:**

The reviewers generally saw the merits of the proposal. There was general interest in the paper. The authors have responded to several issues raised by the reviewers.  On reading the paper at a fairly high-level, it does appear interesting and novel. The authors in the rebuttal have clarified some of the issues raised by the reviewers. I hope these clarifications will be directly incorporated in the final version of the paper. I suggest borderline acceptance.

**Additional Comments On Reviewer Discussion:**

The reviewer who had the negative review did not quite respond to the authors. The authors gave a detailed explanation for the issues raised but the reviewer did not respond to those rebuttals as well. Hence I down weighted that reviewer's rating when recommending acceptance.

---

### Decision · Program_Chairs · 2025-01-22

Accept (Poster)